# Dust-wind interactions can intensify aerosol pollution over eastern China

Yang Yang[1,2], Lynn M. Russell[1], Sijia Lou[1,2], Hong Liao[3], Jianping Guo[4], Ying Liu[2], Balwinder Singh[2] & Steven J. Ghan[2]

Eastern China has experienced severe and persistent winter haze episodes in recent years due to intensification of aerosol pollution. In addition to anthropogenic emissions, the winter aerosol pollution over eastern China is associated with unusual meteorological conditions, including weaker wind speeds. Here we show, based on model simulations, that during years with decreased wind speed, large decreases in dust emissions (29%) moderate the wintertime land–sea surface air temperature difference and further decrease winds by $-0.06 \, (\pm 0.05) \, \mathrm{m \, s^{-1}}$ averaged over eastern China. The dust-induced lower winds enhance stagnation of air and account for about 13% of increasing aerosol concentrations over eastern China. Although recent increases in anthropogenic emissions are the main factor causing haze over eastern China, we conclude that natural emissions also exert a significant influence on the increases in wintertime aerosol concentrations, with important implications that need to be taken into account by air quality studies.

[1] Scripps Institution of Oceanography, University of California, San Diego, La Jolla, California 92093, USA. [2] Atmospheric Science and Global Change Division, Pacific Northwest National Laboratory, Richland, Washington 99354, USA. [3] School of Environmental Science and Engineering/Joint International Research Laboratory of Climate and Environment Change, Nanjing University of Information Science & Technology, Nanjing 210044, China. [4] State Key Laboratory of Severe Weather, Chinese Academy of Meteorological Sciences, Beijing 100081, China. Correspondence and requests for materials should be addressed to L.M.R. (email: lmrussell@ucsd.edu)

In recent years, eastern China has suffered from heavy haze events with high aerosol concentrations, which has adverse impacts on hundreds of millions of people across China[1-3]. Aerosol pollution harms health by transporting and delivering hazardous substances into lung tissue and bloodstreams[4]. In addition, the intensification of aerosol pollution perturbs the Earth's energy balance and affects regional climate over eastern China[5,6].

Natural aerosols are particularly important for the global energy balance due to their large contribution to the total aerosol loading[7]. They influence the variability of cloud radiative effects[8] and cause uncertainty in aerosol–cloud–precipitation interactions[9,10]. Dust is one of the most important natural aerosols, which affects the Earth's radiative balance mainly through scattering and absorbing solar and thermal radiation[11,12] and influences the vertical profile of the atmospheric heating rate[13] and monsoon precipitation[14].

East Asia is one of the largest sources of dust emissions (after North Africa). Source regions of dust over East Asia include the Gobi Desert and highlands of northwestern China[15]. Interannual variations in dust emissions are large[16] and may influence anthropogenic aerosol pollution over eastern China through changing meteorological fields. However, few previous studies have examined the effects of interannual variability of dust on meteorological fields, especially during winter when aerosol pollution is the most severe over eastern China.

In recent years, the impacts of climate change on aerosol pollution have received increasing attention[17,18]. Aerosol pollution events over eastern China have been attributed to changes in meteorological fields and climate indices associated with climate change, such as Arctic sea ice[19], local precipitation, surface wind[20] and atmospheric circulation[21,22]. Wind speed has been found to be one of the most important factors affecting haze over eastern China[23,24]. Weaker wind speed in the lower layer of the atmosphere reduces aerosol transport/dispersion, favours the accumulation of aerosols near their sources and increases local aerosol concentrations. Here we perform two 150-year simulations at preindustrial conditions, with (referred to as IRUN) and without (referred to as DRUN) interannual variations in dust emissions using the Community Earth System Model (CESM, see Methods section). With these simulations, we suggest that dust intensifies wintertime anthropogenic aerosol pollution over eastern China by reducing wind speed.

## Results

**Decreased dust during weak winds.** During the East Asian winter monsoon season, strong northwesterly winds prevail in northern China and weak northeasterly winds spread over southern China and the South China Sea (see Supplementary Fig. 1a). Compared to normal conditions, the wind anomalies during weak wind conditions (years with wind speed lower than the 10th percentile, see Methods) are southeasterly and south-westerly over northern and southern China (vectors in Fig. 1a and Supplementary Fig. 1b), respectively. These anomalous winds during weak wind conditions prevent the transport of aerosols by northwesterly and northeasterly winds over northern and southern China, respectively, and inhibit the dispersion of pollutants over eastern China. The model captures well both the climatological mean wind fields (Supplementary Fig. 1c,e) and the wind anomalies between weak wind and normal conditions (Supplementary Fig. 1d,f) over eastern China compared to those derived from the GEOS-4 assimilated meteorological fields and NCEP/NCAR (National Centers for Environmental Prediction/National Center for Atmospheric Research) reanalysis data. Since the Gobi Desert (100–110° E, 35–45° N) borders on eastern

China (110–122.5° E, 20–45° N), wind speed over the Gobi Desert region shows similar temporal variability to that over eastern China. The correlation coefficients between wind speed over the Gobi Desert and spatially averaged wind speed over eastern China are $+0.4 \sim +0.8$ in the IRUN simulation, GEOS-4 assimilated meteorological fields and NCEP/NCAR reanalysis data (Supplementary Fig. 2a–c). This similarity of wind speed variability leads to the co-variation of dust emissions over the Gobi Desert and wind speed over eastern China (Supplementary Fig. 2d). Therefore, the anomalous decreasing wind speeds during weak wind conditions lead to decreasing dust emissions (contours in Fig. 1a). Over the Gobi Desert, the simulated December–January–February (DJF) dust emission is 8.1 Tg in the weak wind conditions, that is, 29% lower than the 11.4 Tg characteristic of the normal conditions. The highlands of north-western China only show a small change in dust emissions during these years because this region is affected by different wind conditions than those in eastern China. These changes in dust emissions between weak wind and normal conditions produce changes in dust concentrations.

The variations in dust emissions lead to decreased dust column burden in weak wind conditions compared to those in normal conditions, with an average burden anomaly of $-6.7 \, \text{mg m}^{-2}$ over eastern China (Fig. 1b). The maximum decreases are located over northern China (110–122.5° E, 30–45° N), with simulated values of $-15$ to $-20 \, \text{mg m}^{-2}$, in contrast to southern China (110–122.5° E, 20–30° N) and the South China Sea where the changes are within $\pm 5 \, \text{mg m}^{-2}$. Effects from other factors contributing to the difference such as transport and (largely wet) deposition are removed by subtracting the DRUN results from the IRUN values. Therefore, these changes in dust burden are only due to changes in dust emissions between weak wind and normal conditions.

The simulated difference in the dust column burden between weak wind and normal conditions (Fig. 1c) is similar to the dust column burden difference resulting from the dust emission variability (Fig. 1b) in both spatial pattern and magnitude, suggesting that emission variability is the main factor that decreases dust concentrations in weak wind conditions over eastern China. The observed Aerosol Index, which is a measure of absorbing aerosols in the lower troposphere, also decreases over eastern China during weak wind conditions compared to normal conditions for the years of 1987–1992 (Fig. 1d). This is consistent with the decrease in dust emissions and concentrations during weak wind conditions in the IRUN simulation. The surface dust storm observations also show lower dust storm frequency over a majority of sites in both the Gobi Desert and northern China during weak wind conditions in DJF compared to normal conditions (Supplementary Fig. 3), partly supporting the simulated decrease in simulated airborne dust. In addition, both the simulated dust aerosol optical depth (AOD) from the 150-year IRUN simulation and the coarse AOD from Moderate Resolution Imaging Spectroradiometer aerosol products of the Terra satellite for 2001–2016 over the downwind regions of the East China Sea correlate positively with wind speed averaged over eastern China (Supplementary Fig. 4). However, the spatial distributions of the positive correlations show some differences between the model simulations and satellite records, probably due to the uncertainties associated with the aerosol parameterizations in the model, cloud contamination, retrieval bias and signal uncertainties in the satellite data[25]. In addition, coarse AOD might not be dominated by dust particles. Nonetheless, these observational records support the results that both dust emissions at the source of the Gobi Desert and airborne dust over eastern China decrease during weak wind conditions.

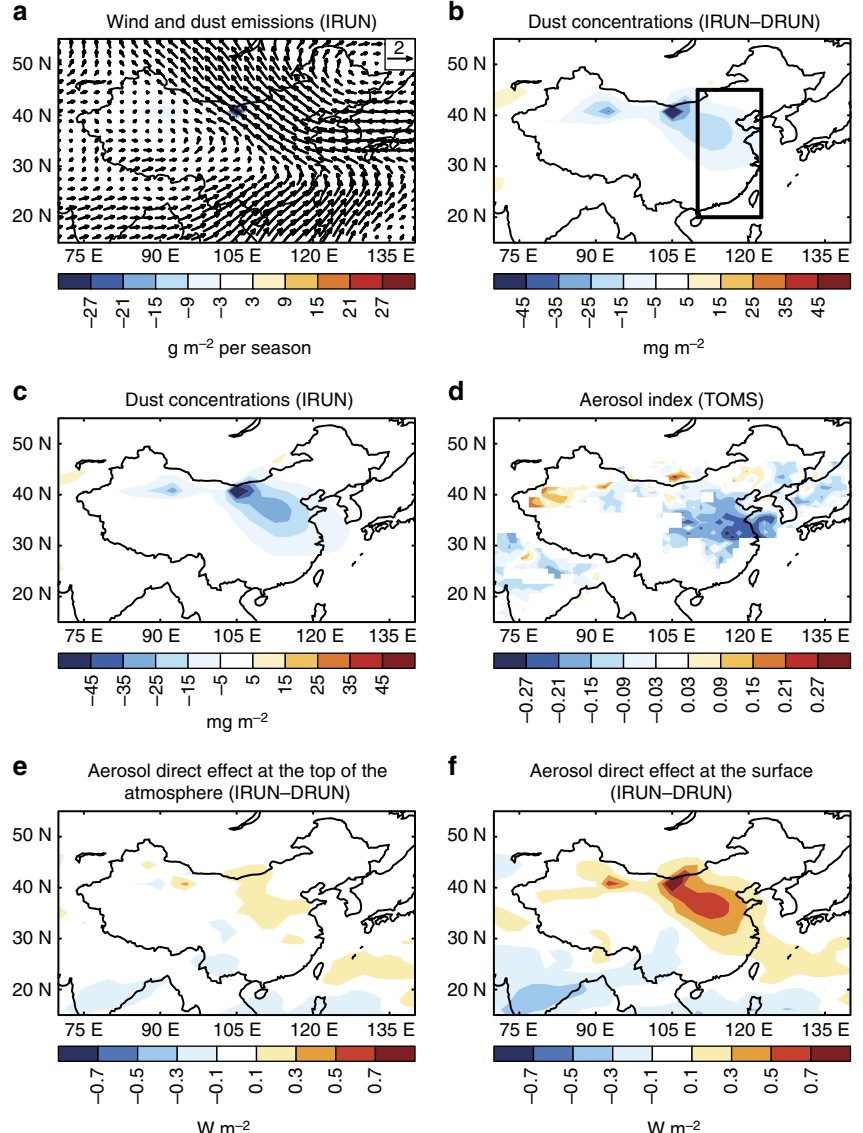

**Figure 1 | Differences in simulated wind and dust in China at preindustrial conditions.** (**a**) Composite differences in wind field at 850 hPa (vectors, unit: m s$^{-1}$) and dust emissions (contours, unit: g m$^{-2}$ season$^{-1}$) between weak wind and normal conditions in the standard simulation with interannual variations in dust emissions (IRUN), which are calculated by $V_{Weak, IRUN} - V_{Normal, IRUN}$. $V$ represents variables. (**b**) Changes in dust column burden (unit: mg m$^{-2}$) between weak wind and normal conditions due to the interannual variations in dust emissions, which are calculated by $(V_{Weak, IRUN} - V_{Normal, IRUN}) - (V_{Weak, DRUN} - V_{Normal, DRUN})$ based on simulations with (IRUN) and without (DRUN) interannual variations in dust emissions. (**c**) Composite differences in dust column burden (unit: mg m$^{-2}$) between weak wind and normal conditions in the IRUN simulation. (**d**) Composite differences in Total Ozone Mapping Spectrometer (TOMS) Aerosol Index between weak wind and normal conditions calculated based on 850 hPa wind speed from NCEP/NCAR meteorological fields for 1979–1993. Changes in shortwave aerosol direct radiative effect (unit: W m$^{-2}$) (**e**) at the TOA and (**f**) the surface, respectively, between weak wind and normal conditions due to the interannual variations in dust emissions. The region boxed in (**b**) is used to represent eastern China (110–122.5° E, 20–45° N).

Because of the decrease in dust column burden, the radiative cooling effects of aerosols, calculated as changes in simulated shortwave radiative flux with and without scattering and absorption by aerosols (see Methods section), decrease both at the top of the atmosphere (TOA) and at the surface in weak wind conditions compared to normal conditions, leading to the anomalous warming of $+0.06$ and $+0.20$ W m$^{-2}$, respectively, over eastern China (Fig. 1e,f). The greater warming at the surface than that at TOA results in anomalous cooling in the atmosphere, especially over northern China. The heating at the surface and cooling in the atmosphere could further destabilize the atmosphere and enhance convection. Although the decrease in dust also leads to anomalous cooling in the longwave bands, its influence is much smaller compared to the change in the shortwave bands, with a magnitude of anomalous cooling of $-0.01$ W m$^{-2}$ at TOA and $-0.07$ W m$^{-2}$ at the surface averaged over eastern China (Supplementary Fig. 5).

**Decreased dust reinforces weakening of winds.** Between 35° N and 45° N over eastern China, decreases in dust emission lead to anomalous heating below 900 hPa (Fig. 2a), which increases the heating rate in the lower atmosphere by vertical diffusion. The simulated heating rates are larger than $+0.4$ K per day in weak wind conditions compared to normal conditions, which produces an anomalous ascent around 35° N over eastern China (Fig. 2b).

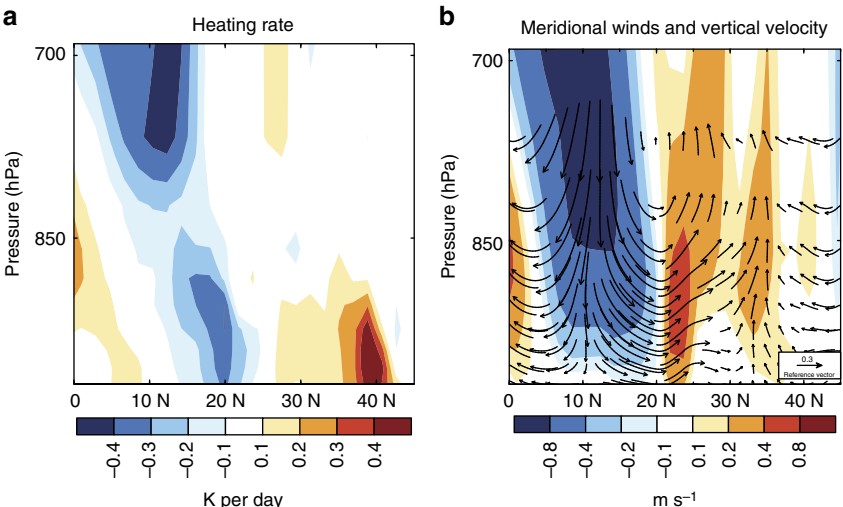

**Figure 2 | Dust-induced changes in vertical profiles of heating rate and winds.** Changes in (**a**) atmospheric heating rate (unit: K per day) and (**b**) meridional wind (vectors, unit: m s$^{-1}$) and vertical velocity (contours, unit: Pa s$^{-1}$) scaled by a factor of $-100$ averaged over 110–122.5° E between weak wind and normal conditions due to the interannual variations in dust emissions.

This anomalous ascent over land partially offsets the land–sea circulation in winter over eastern China, reducing the subsidence over land and the ascent over ocean and further decreasing the heating of the atmosphere over the South China Sea. This anomalous heating over land and cooling over ocean decreases the wintertime land–sea surface air temperature difference between eastern China and the South China Sea, which leads to an anomalous southerly wind in the lower layer of the atmosphere between 10° N and 35° N.

The decreases in dust emissions during weak wind conditions produce anomalous southwesterly winds over southern China (Fig. 3a), further offsetting northeasterly winds in normal conditions and weakening wind speed in weak wind conditions, with a statistically significant value of $-0.06$ ($\pm 0.05$) m s$^{-1}$ averaged over eastern China. We also examined the effect of dust emission variability on weakening wind speed by looking at the weak wind conditions defined as years with wind speed lower than the 20th and 30th percentiles (Supplementary Fig. 6). Changes in wind direction using 20th and 30th percentile thresholds are almost the same as that of the 10th percentile threshold, but their magnitudes are reduced from $-0.06$ ($\pm 0.05$) m s$^{-1}$ for the 10th percentile threshold to insignificant values of $-0.02$ ($\pm 0.02$) and $-0.01$ ($\pm 0.01$) m s$^{-1}$ for the 20th and 30th percentile thresholds, respectively, averaged over eastern China. This indicates that effects of dust interannual variability have higher effects on weaker wind speed conditions than in normal conditions in winter over eastern China.

The interactions between dust emissions and wind speed over eastern China produce a positive dust–wind feedback (Fig. 3b). First, dust emissions are lower over the Gobi Desert during weak wind conditions, leading to lower dust concentrations and a positive anomaly in aerosol direct radiative effects over northern China. This translates into anomalous heating over northern China and decreases the land–sea surface air temperature difference between eastern China and the South China Sea, which further reinforces the weakening of winds.

**Feedback between dust and wind intensifies pollution.** A simulation of modern conditions with the GEOS-Chem model (see Methods section) is used to confirm the negative correlation between aerosol concentration and wind speed and to quantify the impact of dust-induced decrease in wind speed on aerosol

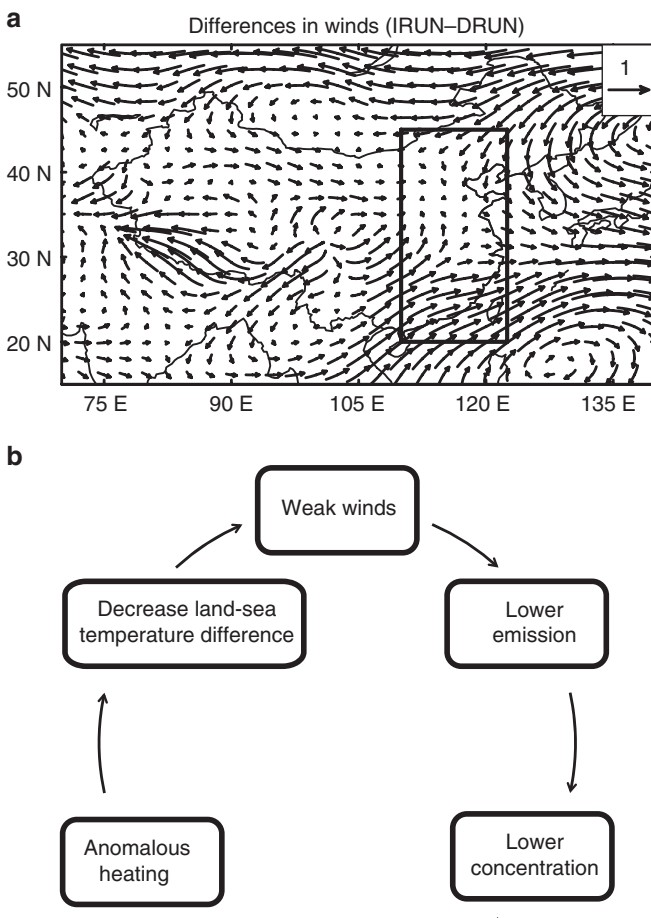

**Figure 3 | Changes in horizontal winds and dust–wind feedback.** (**a**) Changes in wind fields (unit: mg m$^{-2}$) between weak wind and normal conditions due to the interannual variations in dust emissions. (**b**) The dust–wind feedback over eastern China. The region boxed in (**a**) is used to represent eastern China (110–122.5° E, 20–45° N).

pollution over eastern China. Statistically significant negative regression coefficients between the 850 hPa wind speed averaged over eastern China and the simulated surface layer $PM_{2.5}$ (particle mass less than 2.5 μm in diameter) concentrations in DJF spread over almost all of eastern China, with the maximum magnitude exceeding $-9\,\mu g\,m^{-3}\,(m\,s^{-1})^{-1}$ between 30° N and 40° N (Fig. 4a). The negative coefficients indicate that weak wind speeds tend to induce air stagnation and aerosol accumulation in the lower atmosphere over eastern China, with high $PM_{2.5}$ concentrations in weak wind conditions.

The spatially averaged $PM_{2.5}$ concentrations have a strong negative correlation with wind speed over eastern China, with a statistically significant correlation coefficient of $-0.70$ (Fig. 4b). The linear regression coefficient between $PM_{2.5}$ concentrations and wind speed in the GEOS-Chem simulation is $-3.83\,\mu g\,m^{-3}\,(m\,s^{-1})^{-1}$.

Therefore, interannual variations in dust emissions weaken wind speed by $-0.06\,m\,s^{-1}$ and enhance stagnation (defined based on wind speed[26]) during weak wind conditions. The enhanced stagnation increases the DJF mean $PM_{2.5}$ concentrations by $0.23\,\mu g\,m^{-3}$, about 13% of the total increase in $PM_{2.5}$ concentrations between weak wind (years 1990 and 1995) and normal conditions (1986–2006) in the GEOS-Chem simulation.

**Observational evidence.** Observed daily dust storm events and daily atmospheric visibility measurements in China, together with the lagged correlation analysis (Fig. 5), provide support for the finding that the dust–wind feedback enhances aerosol pollution over eastern China. When lagging is not applied, dust storm occurrence over the Gobi Desert and atmospheric visibility over the dust source region and the downwind northern China are negatively correlated. This result suggests that, when dust storms occurred in association with strong winds, the visibility decreased significantly over these regions on the same day as the dust storms. One day after the dust storms, the visibility increased due to the decreased dust aerosol concentration, with 61 of the 154 sites over eastern China showing statistically significant positive correlations over eastern China. Lags of 2 and 3 days showed that the visibility increased substantially in eastern China, with statistically significant positive correlations at 91 and 92 sites, respectively. If the sudden increases in positive correlations at 2–3 days were only due to the dust-induced visibility changes, the correlations would not have large changes after 3 days. However, these large positive correlations decreased substantially at lags of 4 and 5 days, with only 79 sites showing statistically significant positive correlations over eastern China. This result indicates that a meteorological response to the Gobi dust variability exists with 2–3 day lag times, leading to decreases (increases) in visibility and increases (decreases) in aerosol concentrations over eastern China when there are lower (higher) dust emissions over the Gobi Desert. This lagged correlation analysis of daily data strongly support our finding that decreases in dust emissions reinforce weakening of wind speed and intensify aerosol pollution over eastern China.

In addition, the weaker wind speed induced by the dust–wind feedback simulated in this study is stronger over southern China than over northern China (Fig. 3a). The observed haze days and dust storm frequency averaged over the Gobi Desert in DJF also show a stronger negative correlation over southern China than over northern China (Supplementary Fig. 7a). The decrease in surface net shortwave flux derived from the Clouds and the Earth's Radiant Energy System (CERES) is also larger over southern China during weak wind conditions than in normal wind conditions (Supplementary Fig. 7b), even though anthropogenic aerosols are more concentrated over northern China. These results also provide observational support for the contribution of the modeled feedback effects in recent observations, although the haze days and shortwave flux are also influenced by dust variability.

## Discussion

Compared to the internal variability of atmospheric circulation, aerosols play a relatively small role in affecting atmospheric circulation. However, aerosol effects are still important, as they provide understanding of the mechanisms that control atmospheric circulation and its variability. Another example of this type of effect is that radiative forcing of aerosols was reported to account for about 70% of variations in northern tropical Atlantic Ocean temperatures[27] and influence climate variability in the tropical and North Atlantic[28–39]. In addition, aerosols are also found to intensify Arabian Sea tropical cyclones[30], enhance storm

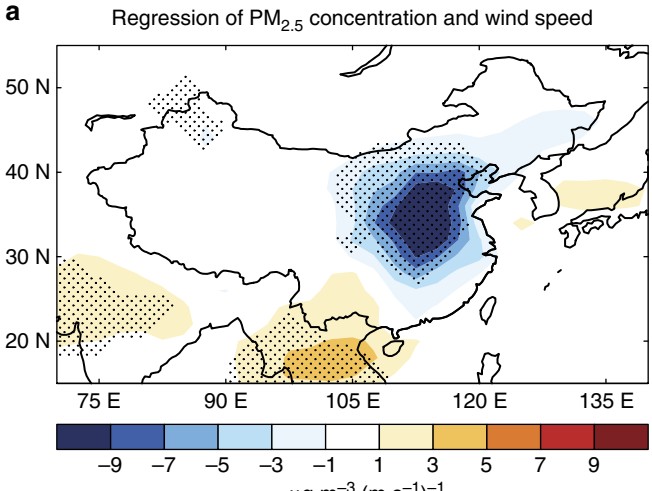

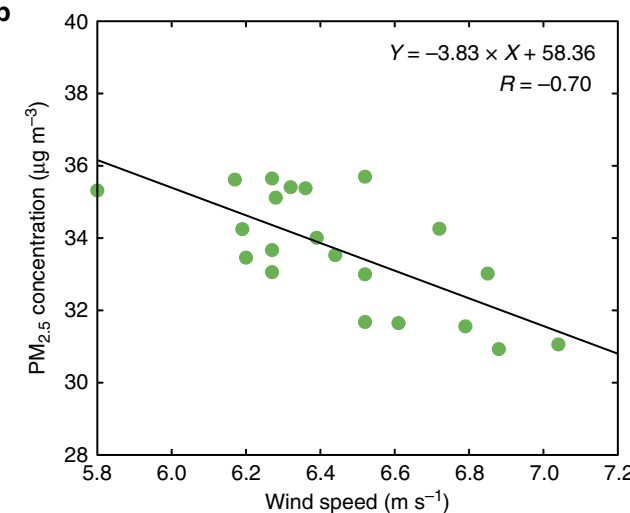

**Figure 4 | Regression of wind and $PM_{2.5}$ in GEOS-Chem model.**
(**a**) Regression coefficient between the 850 hPa wind speed averaged over eastern China (110–122.5° E, 20–45° N) and the simulated surface layer $PM_{2.5}$ concentrations in winter (unit: $\mu g\,m^{-3}\,(m\,s^{-1})^{-1}$). The dotted areas indicate statistical significance with 95% confidence. (**b**) Scatter plot between the area-averaged 850 hPa wind speed and area-averaged surface layer $PM_{2.5}$ concentrations over eastern China during 1986–2006. Wind speed is derived from GEOS-4 meteorological fields. $PM_{2.5}$ concentrations are from the GEOS-Chem model simulation. Regression model and correlation coefficient between area-averaged wind speed and $PM_{2.5}$ concentrations are shown on top right.

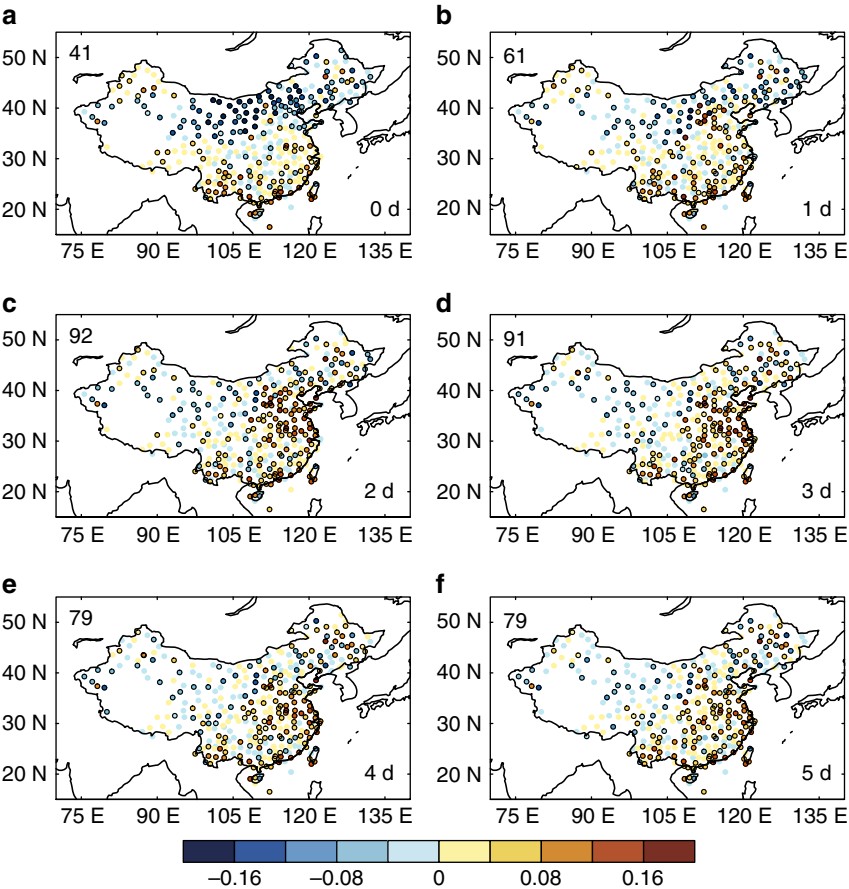

**Figure 5 | Lagged correlation between observed dust and visibility.** Lagged correlation coefficients between observed daily dust storm events averaged over the Gobi Desert region and atmospheric visibility over 346 meteorological stations in China in DJF for years 1981–2015. Sites with outlines indicate statistical significance with 95% confidence. The lag time for the atmospheric visibility relative to the leading dust storm events are (**a**) 0, (**b**) 1, (**c**) 2, (**d**) 3, (**e**) 4 and (**f**) 5 days, as shown at the bottom right of each panel. The number of stations with statistically significant positive correlations over eastern China is shown at the top left of each panel. Records from a total of 154 stations located in eastern China were used for this analysis.

track intensity[31], modulate mid-latitude cyclones[32] and weaken the South Asian summer monsoon[33]. Therefore, although aerosol effects are relatively small, they are important for understanding variability in atmospheric circulation.

Since haze episodes occur on a daily timescale, it is important to consider not just the seasonal mean increase but the change in daily maximum $PM_{2.5}$ concentrations. Although the absolute value of dust-induced change in seasonal mean $PM_{2.5}$ concentrations is relatively small, the daily change should be much larger due to the stronger wind variability and the more effective role of dust in weaker wind speed conditions. For example, the DJF mean wind speed at 850 hPa ranges from 5.8 to 7.4 m s$^{-1}$ in the 21-year GEOS-4 assimilated meteorological field, whereas daily wind speed has much larger variability ranging from 2.4 to 14.5 m s$^{-1}$. The results in this study use the perspective of the climatological mean by incorporating monthly averaged model simulations. On shorter timescales, the transport pathways and spatial extent of individual dust events would be much more variable locally.

In this study, the interactions between dust emissions and wind speeds were simulated using the CESM model but the present-day $PM_{2.5}$ and their relationships to wind speed were quantified by GEOS-Chem model simulations. Considerable uncertainties exist in simulations of dust emissions as well as in their transport and other interactions with meteorology in modeling simulations. For this reason, different models may provide different estimates of the specific magnitude of the dust–wind feedback. However, we

expect that the basic physics that drive these results will provide the same type of feedback effect in all models that include wind-dependent dust emission parameterizations and aerosol radiative forcing effects on circulation. In addition, the 13% of the total increase resulting from dust–wind interaction is from a simulation that can overestimate or overlook many factors. Although observational data in this study also imply the existence of dust–wind feedback, the magnitude of the influence may differ from that in model simulations.

In addition to emissions from natural sources, atmospheric dust could also originate from soils disturbed by human activities (so called anthropogenic dust), which account for a substantial amount of dust loading over northern China[34]. The model used in this study does not separately include anthropogenic dust sources, although changes in wind speed would also influence the emissions and concentrations of this dust, further perturbing meteorological fields. In addition, airborne dust was also found to directly contribute to wintertime regional haze over eastern China from satellite observations[35].

In addition to winds, aerosols could also affect other meteorological fields, such as planetary boundary layer height, which also enhances the concentration of aerosol pollution over China[36,37]. Natural aerosols, such as dust, may also influence the planetary boundary layer and then affect aerosol pollution. In turn, anthropogenic activities also perturb dust concentration[38], which might further affect anthropogenic aerosols. These effects require more simulations, such as with and without dust or anthropogenic

emissions in the model, and need to be examined in future studies. Also, increasing anthropogenic aerosols can induce a more stable atmosphere, which leads to accumulation of air pollutants and contributes to haze formation over eastern China[39].

Aerosol pollution is one of the most important environmental issues in China. We have shown that decreases in dust emissions weaken wind speed when the wind speed is lower than the 10th percentile and account for 13% of the increases in simulated $PM_{2.5}$ aerosol concentrations. The results presented in this study highlight the influence of natural aerosols on the wintertime anthropogenic aerosol pollution over eastern China. Considering the weakening of East Asian monsoon winds[40] and the decrease in dust days over the Gobi Desert[41] in the past few decades, our results could be contributing to the increasing haze episodes over eastern China, as well as the underestimation of modelled aerosol peak mass concentrations in haze events[42].

## Methods

**CESM model and simulations.** To quantify the influence of interannual variations of dust emissions on wintertime wind speed over eastern China, we have performed two 150-year simulations at preindustrial conditions, with (referred to as IRUN) and without (referred to as DRUN) interannual variations in dust emissions using the CESM version 1.2.1 (ref. 43). The atmospheric model used here is version 5 of the Community Atmosphere Model (CAM5), with resolution of 1.9° latitude by 2.5° longitude and 30 vertical layers ranging from the surface to 3.6 hPa. The CESM-CAM5 treats the properties and processes of major aerosol components (sea salt, mineral dust, sulfate, black carbon, primary organic matter and secondary organic aerosol) in the modal aerosol module (MAM3) (ref. 44). Aerosol size distributions are represented by three lognormal modes: Aitken, accumulation, and coarse modes. Mass mixing ratios of different aerosol species and the number mixing ratio are predicted for each mode. Aerosol optical properties are parameterized in the model[45]. Dust emissions and meteorological fields are calculated online. Dust is emitted in both accumulation and coarse modes[46]. The modal aerosol representation in CESM has been evaluated in detail in a previous study[44]. In the CESM, the diagnostic radiative fluxes can be calculated both with and without aerosol scattering and absorption. The aerosol direct radiative effect could be calculated from the differences of these two sets of radiative flux outputs. Two 150-year simulations are performed. The first is the standard simulation of preindustrial conditions using interactive emissions (IRUN). In this simulation, emissions of dust are driven by the meteorological fields, which vary year to year. The second is a sensitivity simulation of preindustrial conditions using prescribed emissions of dust (DRUN). The dust emissions in this simulation are interpolated in time between the 12 monthly mean values derived from the 150-year IRUN simulation and do not change with the actual yearly values of meteorological fields. Hence, this simulation contains no interannual variability of dust emissions. The model setup is otherwise the same as that in IRUN.

All anthropogenic and biomass burning emissions are fixed at the 1850 level in the CESM simulations to focus on dust effects on wind speed.

**GEOS-Chem model and simulations.** To estimate the effects of dust-induced changes in wind speed on the wintertime $PM_{2.5}$ (sum of sulfate, nitrate, ammonium, black carbon and organic carbon aerosols) concentrations over eastern China, aerosol concentrations are simulated in a 21-year GEOS-Chem simulation version 8.02.01 in winter for recent modern conditions driven by the GEOS-4 assimilated meteorological fields from December 1985 to March 2006 and fixed anthropogenic and biomass burning emissions in 2005 to focus on the effects of meteorology on aerosol pollution. The version of the model used here has a horizontal resolution of 2° latitude by 2.5° longitude with 30 vertical layers from surface up to 0.01 hPa. The GEOS-Chem model has fully coupled $O_3$-$NO_x$-hydrocarbon chemistry and aerosols including sulfate, nitrate, ammonium, organic carbon, black carbon, mineral dust and sea salt. The detailed description of aerosol in the model can be found at http://acmg.seas.harvard.edu/geos. The performance of GEOS-Chem model in simulating aerosols over eastern China has been evaluated in many previous studies[47–49].

**Weak wind conditions.** Aerosol pollution episodes tend to occur during periods with weak wind speeds in winter over eastern China. In this study, we define weak wind conditions as years with wind speed lower than the 10th percentile of 150-year climatological DJF mean wind speed at 850 hPa averaged over eastern China (110–122.5° E, 20–45° N). Fifteen years are identified as weak wind conditions based on the 10th percentile of wind speed as the threshold for the 150-year CESM simulations. Variables for normal conditions are calculated as the 150-year climatological mean in DJF. Significance levels are determined according to the Wilcoxon rank-sum test. To quantify the influence of dust emission variability on

meteorological fields, such as heating rate and winds, $(V_{Weak, IRUN} - V_{Normal, IRUN}) - (V_{Weak, DRUN} - V_{Normal, DRUN})$ is calculated in this study to isolate the influence of feedbacks on dust emissions from changing wind speed (where $V$ represents the variable of interest).

**Data availability.** Daily dust storm frequency over 753 Chinese meteorological sites for 1981–2015 was provided by the National Meteorological Information Center, China Meteorological Administration (http://portal.nersc.gov/project/m1374/Dust_-Wind). Daily atmospheric visibility data collected from 346 meteorological stations in China for 1981–2015 were from National Climatic Data Center (NCDC) Global Summary of Day (GSOD) database (http://www7.ncdc.noaa.gov/CDO/cdoselect.cmd?datasetabbv=GSOD&countryabbv=&georegionabbv=). Total Ozone Mapping Spectrometer (TOMS) daily Level-2 Aerosol Index for years of 1979–1993 was from http://disc.sci.gsfc.nasa.gov/data-holdings/PIP/aerosol_index.shtml. AOD data for 2001–2016 were from Moderate Resolution Imaging Spectroradiometer (MODIS) Level-3 aerosol products of the Terra satellite (https://ladsweb.nascom.nasa.gov). Surface net shortwave flux was from the CERES-EBAF Level-3B data set for years 2001–2016 (https://ceres.larc.nasa.gov/order_data.php). NCEP/NCAR (National Centers for Environmental Prediction/National Center for Atmospheric Research) reanalysis data from 1979 to 2016 were from http://www.esrl.noaa.gov/psd/data/gridded/data.ncep.reanalysis.html. The data and codes for these results are posted at: http://portal.nersc.gov/project/m1374/Dust_Wind. All the figures are created by the National Center for Atmospheric Research Command Language (https://www.ncl.ucar.edu).

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

## Acknowledgements

This research was supported by the National Science Foundation grant AGS-1048995 and by DOE DE-SC0006679 as part of the U.S. Department of Energy, Office of Science, Biological and Environmental Research, Decadal and Regional Climate Prediction using Earth System Models (EaSM) program. The Pacific Northwest National Laboratory is operated for the DOE by Battelle Memorial Institute under contract DE-AC05-76RLO 1830. The National Energy Research Scientific Computing Center (NERSC) provided computational resources. We acknowledge support from the DOE, Office of Science, Biological and Environmental Research as part of the Regional and Global Climate Modeling program.

## Author contributions

L.M.R. and S.J.G. designed the research; Y.L., B.S. and S.J.G. performed the CESM model simulations; H.L. provided the GEOS-Chem model simulations; Y.Y., S.L. and J.G. analysed the data. All the authors discussed the results and wrote the paper.

## Additional information

**Competing interests:** The authors declare no competing financial interests.

**How to cite this article**: Yang, Y. *et al.* Dust–wind interactions can intensify aerosol pollution over eastern China. *Nat. Commun.* **8**, 15333 doi: 10.1038/ncomms15333 (2017).

**Publisher's note**: 

