## [Peer Review File · Nature Communications]

Reviewers' comments:

Reviewer #1 (Remarks to the Author):

Overall summary:

Based on model simulation, authors find a positive dust-wind feedback, which exert a significant influence on the aerosol pollution besides anthropogenic emission over eastern China. The feedback can be described as weak wind conditions → lower dust emissions → lower dust concentrations → positive anomaly in aerosol direct radiative effects over land → decreases the land-sea surface air temperature difference → further reinforces the weakening of winds. This work might provide a valuable insight that severe winter haze episodes in eastern China contributed by not only anthropogenic factors but also natural factors. The feedback loop is the key mechanism to get the conclusion that dust-wind interactions intensify aerosol pollution over eastern China, while there are still a few questions to be addressed before linking the whole feedback loop.

Major Comment:

1. The definition of weak wind condition in this manuscript is by using the wind speed in 850hPa averaged over eastern China (110–122.5°E, 20–45°N). As we know that the dust source for eastern China is from the western and northern China, how could authors link the dust emissions in western and northern China to wind speed over eastern China? Thus, it is not surprising that the change of dust emissions over northwestern China is small (see Line 86-89 or Fig. 1a).

2. This article designs two experiments, one is with meteorology-dust interactive emissions (IRUN) and other is with prescribed dust emissions from IRUN (NRUN). In NRUN, dust emissions do not change with actual meteorological fields, then would not obviously vary with weak wind and normal wind condition. Therefore, the pattern of changes in dust column burden in Fig.1b and Fig.1c should be similar. So why do not simply use the $V_{Weak, IRUN} - V_{Normal, IRUN}$ instead of $(V_{Weak, IRUN} - V_{Normal, IRUN}) - (V_{Weak, DRUN} - V_{Normal, DRUN})$ in following Fig.1e, Fig.1f, Fig.2?

3. Compared to IRUN, NRUN removes the interannual variability, while haze episodes are on daily or weekly time scale. Why do not use a weather forecast model herein?

4. From Fig.1e, Fig.1f and Fig.2, we can see there is a cooling over South China Sea but without corresponding dust loading changes in Fig.1b. Does the radiative effect only respond to dust aerosol? How do any other aerosol species (sea salt, sulfate and so on) change under weak wind and normal condition?

5. Author may need to discuss the potential impact of anthropogenic dust. Please see that "Huang J., J. Liu, B. Chen, and S. L. Nasiri, 2015: Detection of anthropogenic dust using CALIPSO lidar measurements. *Atmospheric Chemistry and Physics*, 15, 11653–11665, doi:10.5194/acp-15-11653-2015."

Specific Comments

1. The abstract part could be improved by adding some quantitative results to supplement the qualitative description.

2. Line 71-Line 76

"Compared to normal conditions, the wind anomalies during weak wind conditions (years with wind speed lower than the 10th percentile, see Methods) are southwesterly and southeasterly over southern and northern China (vectors in Fig. 1a), respectively."

Please add descriptions that explain why this kind of wind direction is unfavorable for the dispersion of pollutants.

3. Line 112-Line 124

In the discussion of the dust radiative effect, only shortwave radiative effect is considered. It is true that the dust's shortwave radiative effect is stronger than its longwave radiative effect. However the radiative effect of dust in the longwave should also be included, so as to justifiably estimate the net radiative effect of dust.

In line 116-117, it says that there is an anomalous cooling in the atmosphere ("The greater warming at the surface than that at TOA results in anomalous cooling in the atmosphere ..."). But in line 122-124 it says that there is an anomalous heating in the atmosphere below 900 hPa ("Between 35°N and 45°N over eastern China, decreases in dust emission lead to anomalous heating below 900 hPa ..."). Are these two statements contradicting with each other?

4. Line 132-Line 134

"The decreases in dust emissions during weak wind conditions produce anomalous strong southwesterly winds over southern China (Fig. 3a), further weakening wind speed in weak wind conditions ... "

It is a bit confusing, why the STRONG southwesterly winds further WEAKENING the wind speed?

5. Why use GEOS-CHEM model rather than continue to use CESM model to estimate the effects of dust-induced changes in wind speed on the wintertime pollutants? As described in Method section, CESM also has its own aerosol module, which makes it capable to do the similar simulation.

6. Line 165-Line 177

The regression correlation coefficient between PM2.5 concentration and wind speed, given by GEOS-CHEM model, and the dust-induced change of wind speed, given by CESM model, are jointly used to estimate the increase of PM2.5 concentration due to weaken wind speed. However, since these two models is different, is it reasonable to use the estimation given by two different models and make estimation? The insufficiency of this approach should be discussed at least.

7. Line 260-262

Please provide the list of these 15 years with weak wind condition. Because it is important to know whether these 15 years are recent.

Reviewer #2 (Remarks to the Author):

This paper tried to argue that dust radiation induced changes in wind speed have significant impact on increase of particle concentration in eastern China. It is a very interesting topic, especially if their conclusion was based on robust evidence. However, this work obviously lacks adequate support. Here are some specific comments for reference:

1 The authors did not give a convincing description of variations of dust activities from an observational view. Dust storms crease in northwestern China, but the airborne dust aerosols may not.

2 There have been so many works concerning the heavy aerosol pollution in eastern China. As shown by the authors, radiative effects induced by their simulated dust variations were so limited compared with that of the thick haze layers in China. Considering that many evident factors have been emphasized in causing air pollution, influence of changes in dust emissions can be slight rather than significant if they existed.

3 There can be more scientific results if the authors paid more attention on changes in dust emissions and corresponding radiative effects with sufficient support.

4 submit to a special journal may be more suitable for this work after a proper improvement.

Reviewer #3 (Remarks to the Author):

Review of "Dust-wind interactions intensify aerosol pollution over eastern China" by Yang Y. et al. (#ID: NCOMMS-16-20004-T)

Summary: The paper revealed the possible interactions between the natural dust emission and wind speed over eastern China, and discussed their roles on the increase aerosol pollution, especially winter hazy pollution in recent decades, using model simulations. I think the issue of this paper is important, and some findings in the paper favor to recognize haze pollution over eastern China thoroughly. Otherwise, I also found some issues in the paper should be revised or addressed further in order to consolidate its conclusions, so I recommend a major revision before this paper can be published.

Major comments:

1. The major flaw of this paper is that the main conclusions are obtained just basing on two sensitivity simulations, with and without inter-annual variations in dust emissions. This is not convincing to me. The relationships among the wind, aerosol pollution, and dust emission have been well revealed by many previous studies from simulations. If this study is limited at the current level, I suggest rejecting because it is just a copy study and there is no novel. Some results from the observations should be added to consolidate the conclusions. Of course, just several sites are needed if the data are limited.
2. In the experiment design (Line 228-237). The initial conditions are not clear in IRUN. This simulation was driven by the dust emission or the meteorological fields? If driven by the dust emission, is it observation or simulation? Is it open or not? If driven by the meteorological fields, which variables are needed to input? Wind or others?
3. In supplementary Figure 3, the authors showed the relationship between the MAM dust storm frequency index and DJF haze days over eastern China, and their correlation coefficient can be up to +0.64 (I think it should be -0.64??). The relationships between wind and DSFI, wind and haze days in China have been well revealed by many previous studies. The relation between the DSFI and haze days is certainly clear. The most important thing here is that the author should give out the contribution rate of the wind speed due to the natural dust emission to the increased haze days.
4. In recent years, the impacts of climate change on aerosol pollution has been taken increasing attentions. I suggest the authors should also give a review in this aspect. I suggest some referees as follow.
Wang, H. J., H. P. Chen, and J. P. Liu, 2015: Arctic sea ice decline intensified haze pollution in eastern China. *Atmos. Oceanic Sci. Lett.*, 8(1), 1-9.
Zhang, Z., X. Zhang, D. Gong, S.-J. Kim, R. Mao, and X. Zhao, 2016: Possible influence of atmospheric circulations on winter haze pollution in the Beijing-Tian-Hebei region, northern China. *Atmos. Chem. Phys.*, 16, 561-571.
Xu, X., T. Zhao, et al., 2016: Climate modulation of the Tibetan Plateau on haze in China. *Atmos. Chem. Phys.*, 16, 1365-1375.
Wang, H. J., and H. P. Chen, 2016: Understanding the recent trend of haze pollution in eastern China: roles of climate change. *Atmos. Chem. Phys.*, 16, 4205-4211.

Responses to Reviewer #1

Overall summary:

Based on model simulation, authors find a positive dust-wind feedback, which exert a significant influence on the aerosol pollution besides anthropogenic emission over eastern China. The feedback can be described as weak wind conditions → lower dust emissions → lower dust concentrations → positive anomaly in aerosol direct radiative effects over land → decreases the land-sea surface air temperature difference → further reinforces the weakening of winds. This work might provide a valuable insight that severe winter haze episodes in eastern China contributed by not only anthropogenic factors but also natural factors. The feedback loop is the key mechanism to get the conclusion that dust-wind interactions intensify aerosol pollution over eastern China, while there are still a few questions to be addressed before linking the whole feedback loop.

Major Comment:

1. The definition of weak wind condition in this manuscript is by using the wind speed in 850hPa averaged over eastern China (110–122.5°E, 20–45°N). As we know that the dust source for eastern China is from the western and northern China, how could authors link the dust emissions in western and northern China to wind speed over eastern China? Thus, it is not surprising that the change of dust emissions over northwestern China is small (see Line 86-89 or Fig. 1a).

Response:

Wind variability in winter over China is associated with the East Asian winter monsoon system. Due to their adjacent locations, wind speed over the Gobi desert (100–110°E, 35–45°N) and eastern China (110–122.5°E, 20–45°N) tend to have similar variability in winter. The temporal correlation coefficients between wind speed at 850 hPa averaged over the Gobi desert and eastern China are +0.68 in the 150-yr model simulation. Dust emissions are largely influenced by wind speed. Therefore, the dust emissions over the Gobi desert are expected to be correlated with the wind speed over eastern China.

We have added Supplementary Fig. 2a, 2b and 2c to show the correlation coefficients between wind speed at 850 hPa in each grid cell and wind speed averaged over eastern China from model simulation, GEOS-4 assimilated meteorological fields, and NCEP/NCAR reanalysis data. Wind speed over the Gobi desert shows a statistically significant positive correlation with wind speed averaged over eastern China. Supplementary Fig. 2d presents the correlation between dust emissions and wind speeds averaged over eastern China in DJF from the IRUN simulation. Over the Gobi desert, due to the

strong correlation of wind speed, the dust emissions also show a statistically significant positive correlation with wind speed averaged over eastern China, which links the dust emissions in northern China to wind speed over eastern China. Over northwestern China, the wind speed does not show a strong correlation with wind speed over eastern China (Supplementary Fig. 2a, 2b and 2c), leading to a weak correlation between dust emissions and eastern China wind speed (Supplementary Fig. 2d) and a small change in dust emissions between weak wind and normal days defined by wind speed over eastern China (Fig. 1a).

We have added a description to link dust emission over the Gobi desert and wind speed over eastern China: “Since the Gobi desert borders on eastern China (110–122.5°E, 20–45°N), wind speed over the Gobi desert region (100–110°E, 35–45°N) shows similar temporal variability to that over eastern China. The correlation coefficients between wind speed over the Gobi desert and spatially averaged wind speed over eastern China are +0.4~+0.8 from the IRUN simulation, GEOS-4 assimilated meteorological fields and NCEP/NCAR reanalysis data (Supplementary Figs. 2a, 2b and 2c). This similarity of wind speed variability leads to the co-variation of dust emissions over the Gobi desert and wind speed over eastern China (Supplementary Fig. 2d).”

In addition, we also added a figure using MODIS coarse AOD from 2001 to 2016 to support our results. Supplementary Fig. 3 shows the correlation coefficients between simulated dust aerosol optical depth and wind speed averaged over eastern China and correlation coefficients between observed coarse AOD from MODIS and wind speed. Both the model simulation and satellite data showed a positive correlation between dust (coarse) AOD and wind speed over the downwind regions of the East China Sea, suggesting that dust (coarse) AOD decreases during weak wind condition. These correlations support the finding that not only dust emissions but also airborne dust decrease over eastern China during weak wind conditions. We have added the description in the manuscript as “Both the simulated dust aerosol optical depth (AOD) from the 150-year IRUN simulation and the coarse AOD from Moderate Resolution Imaging Spectroradiometer (MODIS) aerosol products of the Terra satellite over 2001–2016 over the downwind regions of the East China Sea correlate positively with wind speed averaged over eastern China (Supplementary Fig. 3). This correlation supports the results that both dust emissions at the source of Gobi desert and airborne dust over eastern China decrease during weak wind conditions.”

2. This article designs two experiments, one is with meteorology-dust interactive emissions (IRUN) and other is with prescribed dust emissions from IRUN (NRUN). In NRUN, dust emissions do not change with actual meteorological fields, then would not obviously vary with weak wind and normal wind condition. Therefore, the pattern of changes in dust column burden in Fig.1b and Fig.1c should be similar. So why do not simply use the

$V_{\text{Weak, IRUN}} - V_{\text{Normal, IRUN}}$ instead of $(V_{\text{Weak, IRUN}} - V_{\text{Normal, IRUN}}) - (V_{\text{Weak, DRUN}} - V_{\text{Normal, DRUN}})$ in following Fig.1e, Fig.1f, Fig.2?

Response:

The meteorological fields, such as heating rate and winds in Fig. 2, have internal variations and are different between weak wind and normal conditions, even in the emission prescribed DRUN simulation. When estimating the influence of dust emission variability on these meteorological fields (Fig. 2), simply using the $V_{\text{Weak, IRUN}} - V_{\text{Normal, IRUN}}$ would include the internal variation of these variables. $(V_{\text{Weak, IRUN}} - V_{\text{Normal, IRUN}}) - (V_{\text{Weak, DRUN}} - V_{\text{Normal, DRUN}})$ would isolate the influence of feedbacks on dust emissions from changing wind speed. That is why we used $(V_{\text{Weak, IRUN}} - V_{\text{Normal, IRUN}}) - (V_{\text{Weak, DRUN}} - V_{\text{Normal, DRUN}})$ instead of $V_{\text{Weak, IRUN}} - V_{\text{Normal, IRUN}}$ in our study.

We have expanded the description in the method section to clarify this: “To quantify the influence of dust emission variability on meteorological fields, such as heating rate and winds, $(V_{\text{Weak, IRUN}} - V_{\text{Normal, IRUN}}) - (V_{\text{Weak, DRUN}} - V_{\text{Normal, DRUN}})$ is calculated in this study to isolate the influence of feedbacks on dust emissions from changing wind speed (where V represents the variable of interest).”

3. Compared to IRUN, NRUN removes the interannual variability, while haze episodes are on daily or weekly time scale. Why do not use a weather forecast model herein?

Response:

In this study, we introduced a positive dust-wind feedback and then quantified its influence on haze over China. Feedbacks of natural aerosols and meteorological fields are not only over a relatively small domain, but also over a larger, even hemispheric scale. For example, Yang et al. (2016a) found the changes in sea salt emissions could enhance ENSO variability by 10%. However weather forecast models are mainly used for regional scales due to computational limitations.

In addition, we found the CESM model could reproduce well the interactions between natural aerosols and meteorological fields in our previous studies (Lou et al., 2016; Yang et al., 2016a,b,c). To keep consistency, the CESM model is also used here to examine the interactions between dust emissions and wind speed over China.

4. From Fig.1e, Fig.1f and Fig.2, we can see there is a cooling over South China Sea but without corresponding dust loading changes in Fig.1b. Dose the radiative effect only respond to dust aerosol? How do any other aerosol species (sea salt, sulfate and so on) change under weak wind and normal condition?

Response:

Because all the anthropogenic emissions are fixed at the preindustrial level in the CESM simulations, the radiative effects of anthropogenic aerosols

(sulfate, BC, OC) are unlikely to change between weak wind and normal conditions, especially over the South China Sea. Sea salt aerosol could respond to dust-induced wind changes between IRUN and DRUN and slightly perturb radiative flux in Fig. 1e and 1f. However, the small negative radiative effect would not lead to the strong anomalous cooling and subsidence over South China Sea shown in Fig. 2.

Actually, the strong cooling and subsidence over the South China Sea are due to the feedback of decreasing dust-induced heating over northern China. In winter, due to the relatively higher temperatures over the South China Sea and lower temperatures over land in eastern China, atmospheric circulation over eastern China consists of ascent over ocean and subsidence over land, which forms a clockwise circulation (Fig. A). With the strong anomalous heating over northern China caused by decreasing dust during weak wind conditions (Fig. 2a), the anomalous ascent (Fig. 2b) partly offsets the subsidence over northern China (Fig. A) and weakens the clockwise circulation, which leads to subsidence over the South China Sea. The anomalous subsidence over the South China Sea further reduces the energy transport from the ocean surface to the lower atmosphere leading to anomalous cooling.

We have revised the description as “Between 35°N and 45°N over eastern China, decreases in dust emission lead to anomalous heating below 900 hPa (Fig. 2a) because of the strong anomalous heating at the surface, which increases the heating rate in the lower atmosphere by vertical diffusion. The heating rates are larger than $+0.4 \text{ K day}^{-1}$ in weak wind conditions compared to normal conditions, which produces an anomalous ascent around 35°N over eastern China (Fig. 2b). This anomalous ascent over land partially offsets the land-sea circulation in winter over eastern China, reducing the subsidence over land and the ascent over ocean and further decreasing the ascent and heating of the atmosphere over the South China Sea. This anomalous heating over land and cooling over ocean decreases the wintertime land-sea surface air temperature difference between eastern China and the South China Sea, which leads to an anomalous southerly wind in the lower layer of the atmosphere between 10°N and 35°N.”

5. Author may need to discuss the potential impact of anthropogenic dust. Please see that "Huang J., J. Liu, B. Chen, and S. L. Nasiri, 2015: Detection of anthropogenic dust using CALIPSO lidar measurements. *Atmospheric Chemistry and Physics*, 15, 11653–11665, doi:10.5194/acp-15-11653-2015."
Response:

Thanks for the suggestion. We have added the discussion of the potential impact of anthropogenic dust in the discussion section, as “In addition to emissions from natural sources, atmospheric dust could also originate from soils disturbed by human activities (so called anthropogenic dust), which account for a substantial amount of dust loading over northern China (Huang et al., 2015). The model used in this study does not separately include

anthropogenic dust sources, although changes in wind speed would also influence the emissions and concentrations of this dust, further perturbing meteorological fields.”

Specific Comments

1. The abstract part could be improved by adding some quantitative results to supplement the qualitative description.

Response:

Thanks for the suggestion. We have added as “Here we show, based on model simulations, decreases in dust emissions by 29% during years with decreased wind speed moderate the wintertime land-sea surface air temperature difference between eastern China and the South China Sea and further decrease winds by $-0.06 (\pm 0.05)$ m s⁻¹ averaged over eastern China. The dust-induced lower winds enhance stagnation of air and account for about 13% of increasing aerosol concentrations over eastern China.”

2. Line 71-Line 76

“Compared to normal conditions, the wind anomalies during weak wind conditions (years with wind speed lower than the 10th percentile, see Methods) are southwesterly and southeasterly over southern and northern China (vectors in Fig. 1a), respectively.”

Please add descriptions that explain why this kind of wind direction is unfavorable for the dispersion of pollutants.

Response:

We have added the description as “These anomalous southeasterly and southwesterly winds during weak wind conditions prevent the transport of aerosols by northwesterly and northeasterly winds over northern and southern China, respectively, and inhibit the dispersion of pollutants over eastern China.”

3. Line 112-Line 124

In the discussion of the dust radiative effect, only shortwave radiative effect is considered. It is true that the dust’s shortwave radiative effect of is stronger than its longwave radiative effect. However the radiative effect of dust in the longwave should also be included, so as to justifiably estimate the net radiative effect of dust.

Response:

Thank for the suggestion. We have added the changes in longwave aerosol radiative effect (ADE) in Supplementary Fig. 4. Because the dust longwave radiative effect is much smaller than shortwave radiative effect over China (Xia and Zong, 2009), the decrease in dust only leads to an anomalous cooling of -0.01 W m⁻² at TOA and -0.07 W m⁻² at the surface averaged over eastern China, smaller than anomalous warming of $+0.06$ and $+0.20$ W m⁻²

from shortwave bands. We also added description as “Although the decrease in dust also leads to anomalous cooling in the longwave bands, its influence is much smaller compared to the change in the shortwave bands, with a magnitude of anomalous cooling of -0.01 W m^{-2} at TOA and -0.07 W m^{-2} at the surface averaged over eastern China (Supplementary Fig. 4).”

In line 116-117, it says that there is an anomalous cooling in the atmosphere (“The greater warming at the surface than that at TOA results in anomalous cooling in the atmosphere ...”). But in line 122-124 it says that there is an anomalous heating in the atmosphere below 900 hPa (“Between 35°N and 45°N over eastern China, decreases in dust emission lead to anomalous heating below 900 hPa ...”). Are these two statements contradicting with each other?

Response:

Thanks for pointing out it. They are not contradicting. Decreasing dust during weak wind conditions could lead to greater warming at the surface and slight warming at TOA by absorption or scattering as part of its direct radiative effect. The difference between surface and TOA results in a negative radiative flux. However, the heating rate in the atmosphere includes not only the direct heating from radiative effects, but also heating from vertical diffusion and moist processes. The decreasing dust leads to heating of the surface, and then the lower atmosphere below 900 hPa is heated by vertical diffusion from the surface (Fig. B).

We have revised the heating rate description to clarify this point: “Between 35°N and 45°N over eastern China, decreases in dust emission lead to anomalous heating below 900 hPa (Fig. 2a) because of the strong anomalous heating at the surface, which increases the heating rate in the lower atmosphere by vertical diffusion.”

4. Line 132-Line 134

“The decreases in dust emissions during weak wind conditions produce anomalous strong southwesterly winds over southern China (Fig. 3a), further weakening wind speed in weak wind conditions ...”

It is a bit confusing, why the STRONG southwesterly winds further WEAKENING the wind speed?

Response:

In normal conditions, northeasterly winds are found over southern China. During weak wind conditions, northeasterly winds become weak, which means ‘anomalous’ southwesterly winds (weak wind – normal) are over southern China between weak wind and normal conditions. The decreasing dust further produces an anomalous southwesterly wind, which strengthens the anomalous southwesterly winds during weak wind conditions and further offsets the northeasterly winds during normal conditions.

We have revised the description as “The decreases in dust emissions

during weak wind conditions produce anomalous southwesterly winds over southern China (Fig. 3a), further offsetting northeasterly winds in normal conditions and weakening wind speed in weak wind conditions.”

5. Why use GEOS-CHEM model rather than continue to use CESM model to estimate the effects of dust-induced changes in wind speed on the wintertime pollutants? As described in Method section, CESM also has its own aerosol module, which makes it capable to do the similar simulation.

Response:

Yes, CESM has ability to simulate aerosols. However, compared to observations, anthropogenic aerosols over East Asia are largely underestimated by CESM, which is probably due to the large wet removal rate and short lifetime of aerosols in the model (Wang et al., 2011; Liu et al., 2012; Wang et al., 2013). The natural aerosols and their climatic effects are reproduced well (Xu et al., 2015; Lou et al., 2016; Yang et al., 2016a,b,c). As a chemical transport model, GEOS-Chem has a much better performance in simulating anthropogenic aerosols in East Asia, which has been evaluated in many previous studies (Y. Wang et al., 2013; Lou et al., 2014; Yang et al., 2015). In addition, GEOS-Chem is driven by assimilated meteorological fields, which provides a more realistic meteorology compared to present day observations. Therefore, we simulated relationships of dust-wind interactions using CESM but we simulated modern PM_{2.5} using GEOS-Chem.

6. Line 165-Line 177

The regression correlation coefficient between PM_{2.5} concentration and wind speed, given by GEOS-CHEM model, and the dust-induced change of wind speed, given by CESM model, are jointly used to estimate the increase of PM_{2.5} concentration due to weaken wind speed. However, since these two models is different, is it reasonable to use the estimation given by two different models and make estimation? The insufficiency of this approach should be discussed at least.

Response:

Thanks for the suggestion. We have added the caveat in the discussion section as “In this study the interactions between dust emissions and wind speeds were simulated using the CESM model but the present-day PM_{2.5} and their relationships to wind speed were quantified by GEOS-Chem model simulations. While the magnitude of the dust influence on PM_{2.5} concentrations could differ between the two models, but such a difference is unlikely to change the main finding that dust-wind interactions intensify aerosol pollution over eastern China.”

7. Line 260-262

Please provide the list of these 15 years with weak wind condition. Because it is important to know whether these 15 years are recent.

Response:

The CESM simulations were performed in preindustrial conditions. All the 150 years represent 1850 level. The variability in these 150-yr simulations is due to the internal variability of the simulated climate system. Therefore, none of the specific years with weak wind conditions are recent, because they all represent the year 1850. To avoid implying that the simulated years represent specific historical years, we do not provide the model year.

Supplementary Fig 2. Correlation coefficients between wind speed at 850 hPa and wind speed averaged over eastern China (110–122.5°E, 20–45°N) in DJF from (a) the 150-yr IRUN simulation, (b) the GEOS-4 assimilated meteorological fields for 1986–2006, and (c) the NCEP/NCAR reanalysis data for years of 1980–2016 (a). (c) Correlation coefficients between dust emissions and wind speeds averaged over eastern China in DJF from the 150-yr IRUN simulation. The region boxed is used to represent the Gobi desert region (100–110°E, 35–45°N, left box) and eastern China (110–122.5°E, 20–45°N, right box). The dotted areas indicate statistical significance with 95% confidence.

Supplementary Fig 3. (a) Correlation coefficients between simulated dust aerosol optical depth (AOD) at 550 nm and wind speed at 850 hPa averaged over eastern China (110–122.5°E, 20–45°N) in DJF from the 150-yr IRUN simulation. (b) Correlation coefficients between observed coarse AOD at 550 nm and wind speed at 850 hPa averaged over eastern China in DJF between 2001 and 2016. Observed AOD is from Moderate Resolution Imaging Spectroradiometer (MODIS) aerosol products of the Terra satellite. Observed wind speed is obtained from NCEP/NCAR reanalysis data. Coarse mode AOD is calculated from total AOD (which was available) minus the fine mode fraction, which is only available over the ocean.

Supplementary Fig 4. Changes in longwave aerosol direct radiative effect (unit: $W\ m^{-2}$) (e) at the top of the atmosphere and (f) the surface, respectively, between weak wind and normal conditions due to the interannual variations in dust emissions.

Fig. A. DJF mean meridional wind (vectors, unit: m s^{-1}) and vertical velocity (contours, unit: Pa s^{-1}) scaled by a factor of -100 averaged over $110\text{--}122.5^\circ\text{E}$

Fig. B. Changes in atmospheric heating rate from vertical diffusion (unit: K day^{-1}) averaged over $110\text{--}122.5^\circ\text{E}$ between weak wind and normal conditions due to the interannual variations in dust emissions.

References

- Yang, Y., L. Russell, S. Lou, M. Lamjiri, Y. Liu, B. Singh, and S. Ghan (2016a), Changes in sea salt emissions enhance ENSO variability, *Journal of Climate*, doi:10.1175/JCLI-D-16-0237.1.
- Yang, Y., et al. (2016b), Impacts of ENSO events on cloud radiative effects in preindustrial conditions: Changes in cloud fraction and their dependence on interactive aerosol emissions and concentrations, *J. Geophys. Res. Atmos.*, 121, doi:10.1002/2015JD024503.
- Yang, Y., L. M. Russell, S. Lou, Y. Liu, B. Singh, and S. J. Ghan (2016c), Rain-aerosol relationships influenced by wind speed, *Geophys. Res. Lett.*, 43, doi:10.1002/2016GL067770.
- Lou, S., L. M. Russell, Y. Yang, L. Xu, M. A. Lamjiri, M. J. DeFlorio, A. J. Miller, S. J. Ghan, Y. Liu, and B. Singh (2016), Impacts of the East Asian Monsoon on springtime dust concentrations over China, *J. Geophys. Res. Atmos.*, 121, 8137–8152, doi:10.1002/2016JD024758.

- Xu, L., D. W. Pierce, L. M. Russell, A. J. Miller, R. C. J. Somerville, C. H. Twohy, S. J. Ghan, B. Singh, J. Yoon, and P. J. Rasch (2015), Interannual to decadal climate variability of sea salt aerosols in the coupled climate model CESM1.0, *J. Geophys. Res. Atmos.*, 120, 1502–1519, doi:10.1002/2014JD022888.
- Wang, M., S. Ghan, M. Ovchinnikov, X. Liu, R. Easter, E. Kassianov, Y. Qian, and H. Morrison (2011), Aerosol indirect effects in a multi-scale aerosol-climate model PNNL-MMF, *Atmos. Chem. Phys.*, 11, 5431–5455, doi:10.5194/acp-11-5431-2011.
- Wang, H., R. C. Easter, P. J. Rasch, M. Wang, X. Liu, S. J. Ghan, Y. Qian, J.-H. Yoon, P.-L. Ma, and V. Vinoj (2013), Sensitivity of remote aerosol distributions to representation of cloud-aerosol interactions in a global climate model, *Geosci. Model Dev.*, 6, 765–782, doi:10.5194/gmd-6-765-2013.
- Liu, X., et al. (2012), Toward a minimal representation of aerosols in climate models: Description and evaluation in the Community Atmosphere Model CAM5, *Geosci. Model Dev.*, 5, 709–739, doi:10.5194/gmd-5-709-2012.
- Wang, Y., Q. Q. Zhang, K. He, Q. Zhang, and L. Chai (2013), Sulfate-nitrate-ammonium aerosols over China: response to 2000–2015 emission changes of sulfur dioxide, nitrogen oxides, and ammonia, *Atmos. Chem. Phys.*, 13, 2635–2652.
- Lou, S., H. Liao, and B. Zhu (2014), Impacts of aerosols on surface-layer ozone concentrations in China through heterogeneous reactions and changes in photolysis rates, *Atmos. Environ.*, 85, 123–138.
- Yang, Y., H. Liao, and S. Lou (2015), Decadal trend and interannual variation of outflow of aerosols from East Asia: Roles of variations in meteorological parameters and emissions, *Atmos. Environ.*, 100, 141–153.
- Xia, X., and X. Zong (2009), Shortwave versus longwave direct radiative forcing by Taklimakan dust aerosols, *Geophys. Res. Lett.*, 36, L07803, doi:10.1029/2009GL037237.

Responses to Reviewer #2

This paper tried to argue that dust radiation induced changes in wind speed have significant impact on increase of particle concentration in eastern China. It is a very interesting topic, especially if their conclusion was based on robust evidence. However, this work obviously lacks adequate support. Here are some specific comments for reference:

1 The authors did not give a convincing description of variations of dust activities from an observational view. Dust storms crease in northwestern China, but the airborne dust aerosols may not.

Response:

We agree with the reviewer that variations of dust emission in source region may not cause the same variations of airborne dust aerosol. Therefore we showed in Fig. 1d the changes in aerosol index (AI) from Total Ozone Mapping Spectrometer (TOMS) measurements between weak wind and normal conditions. Weak wind conditions are determined by the wind speed from NCEP/NCAR reanalysis data. The AI from observations showed similar decreases as the simulated dust concentrations during weak wind conditions. This supports our result that variations of dust aerosol over eastern China in winter are associated with dust emission variability.

In addition, northwesterly winds are typically over the Gobi desert in winter. The decrease in northwesterly winds not only decreases the dust emissions, but also the transport of dust from the Gobi desert to eastern China, both leading to a decrease in airborne dust aerosols over eastern China. Due to the adjacent location of the Gobi desert and eastern China, the wind speed over the Gobi desert (100–110°E, 35–45°N) and over eastern China (110–122.5°E, 20–45°N) tend to have similar variability in winter. Dust emissions are largely influenced by wind speed. Therefore, the dust emissions over the Gobi desert are correlated to the wind speed over eastern China because of their proximity.

We have added Supplementary Fig. 2a, 2b and 2c to show the correlation coefficients between wind speed at 850 hPa in each grid cell and wind speed averaged over eastern China from model simulation, GEOS-4 assimilated meteorological fields, and NCEP/NCAR reanalysis data. Wind speed over the Gobi desert shows a statistically significant positive correlation with wind speed averaged over eastern China. Supplementary Fig. 2d presents the correlation between dust emissions and wind speeds averaged over eastern China in DJF from the IRUN simulation. Over the Gobi desert, due to the strong correlation of wind speed, the dust emissions also show a statistically significant positive correlation with wind speed averaged over eastern China, which links the dust emissions in northern China to wind speed over eastern China. Over northwestern China, the wind speed does not show a strong correlation with wind speed over eastern China (Supplementary Fig. 2a, 2b and 2c), leading to a weak correlation between dust emissions and eastern

China wind speed (Supplementary Fig. 2d) and a small change in dust emissions between weak wind and normal days defined by wind speed over eastern China (Fig. 1a).

We have added a description to link dust emission over the Gobi desert and wind speed over eastern China: “Since the Gobi desert borders on eastern China (110–122.5°E, 20–45°N), wind speed over the Gobi desert region (100–110°E, 35–45°N) shows similar temporal variability to that over eastern China. The correlation coefficients between wind speed over the Gobi desert and spatially averaged wind speed over eastern China are +0.4~+0.8 from the IRUN simulation, GEOS-4 assimilated meteorological fields and NCEP/NCAR reanalysis data (Supplementary Figs. 2a, 2b and 2c). This similarity of wind speed variability leads to the co-variation of dust emissions over the Gobi desert and wind speed over eastern China (Supplementary Fig. 2d).”

In addition, we also added a figure using MODIS coarse AOD from 2001 to 2016 to support our results. Supplementary Fig. 3 shows the correlation coefficients between simulated dust aerosol optical depth and wind speed averaged over eastern China and correlation coefficients between observed coarse AOD from MODIS and wind speed. Both the model simulation and satellite data showed a positive correlation between dust (coarse) AOD and wind speed over the downwind regions of the East China Sea, suggesting that dust (coarse) AOD decreases during weak wind condition. These correlations support the finding that not only dust emissions but also airborne dust decrease over eastern China during weak wind conditions. We have added the description in the manuscript as “Both the simulated dust aerosol optical depth (AOD) from the 150-year IRUN simulation and the coarse AOD from Moderate Resolution Imaging Spectroradiometer (MODIS) aerosol products of the Terra satellite over 2001–2016 over the downwind regions of the East China Sea correlate positively with wind speed averaged over eastern China (Supplementary Fig. 3). This correlation supports the results that both dust emissions at the source of Gobi desert and airborne dust over eastern China decrease during weak wind conditions.”

2 There have been so many works concerning the heavy aerosol pollution in eastern China. As shown by the authors, radiative effects induced by their simulated dust variations were so limited compared with that of the thick haze layers in China. Considering that many evident factors have been emphasized in causing air pollution, influence of changes in dust emissions can be slight rather than significant if they existed.

Response:

Haze in China is a very popular topic, largely because of its extreme importance for the health of millions of people. Previous studies have found many factors leading to haze. We have added an additional description in the introduction: “In recent years, the impacts of climate change on aerosol pollution have received increasing attention (Dawson et al., 2014; Horton et al.,

2014). Aerosol pollution events over eastern China have been attributed to changes in meteorological fields and climate indices associated with climate change, such as Arctic sea ice (Wang et al., 2015), local precipitation, surface wind (Wang and Chen, 2016), and atmospheric circulation (Xu et al., 2016; Zhang et al., 2016).”

Although numerous studies have focused on haze in China, no study proposed the influence from positive dust-wind feedback, that is, that weak wind decreases dust emissions and decreased dust reinforces the weakening of winds and enhances aerosol pollution over eastern China. In addition, this is the first study that highlights the influence of natural aerosols on the wintertime anthropogenic aerosol pollution over eastern China. Therefore, we believe our study contributes a novel aspect to the existing studies of haze, even though the magnitude of its contribution, as specified in the abstract as 13%, is smaller than some of the other factors, especially the man-made ones.

3 There can be more scientific results if the authors paid more attention on changes in dust emissions and corresponding radiative effects with sufficient support.

Response:

Thanks for the suggestion. Globally the simulated dust burden is 22 Tg in this study, within the range of 14–40 Tg from previous estimates (Takemura et al., 2000; Ginoux et al., 2001; Liao et al., 2004). Over Asia, the dust emission simulated in this study is 540 Tg yr⁻¹, within the range of 27–873 Tg yr⁻¹ from 15 global aerosol models compared in the AeroCom project (Huneeus et al., 2011). Across China, the net dust radiative forcing in April is -9.9 W m^{-2} at the surface similar to -9.3 W m^{-2} estimated by Seinfeld et al. (2004) based on the Asian Pacific Regional Aerosol Characterization Experiment data. Based on these comparisons, the model could well simulate the emission, concentration, and radiative forcing of dust compared to observations, which give us confidence to simulate the changes in these variables with the model.

In addition, we have added the changes in longwave aerosol radiative effect (ADE) in Supplementary Fig 4. We also added description as “Although the decrease in dust also leads to anomalous cooling in the longwave bands, its influence is much smaller compared to the change in the shortwave bands, with a magnitude of anomalous cooling of -0.01 W m^{-2} at TOA and -0.07 W m^{-2} at the surface averaged over eastern China (Supplementary Fig. 4).”

4 submit to a special journal may be more suitable for this work after a proper improvement.

Response:

Eastern China has experienced severe and persistent winter haze episodes in recent years, which affects hundreds of millions of people across China. This is the first study that highlights the influence of natural aerosols on the wintertime anthropogenic aerosol pollution over eastern China. This work

includes analysis of both climate variability and air quality, which are directly linked to the daily life and are appropriate for the diverse readership of Nature Communications.

Supplementary Fig 2. Correlation coefficients between wind speed at 850 hPa and wind speed averaged over eastern China ($110\text{--}122.5^\circ\text{E}$, $20\text{--}45^\circ\text{N}$) in DJF from (a) the 150-yr IRUN simulation, (b) the GEOS-4 assimilated meteorological fields for 1986–2006, and (c) the NCEP/NCAR reanalysis data for years of 1980–2016 (a). (c) Correlation coefficients between dust emissions and wind speeds averaged over eastern China in DJF from the 150-yr IRUN simulation. The region boxed is used to represent the Gobi desert region ($100\text{--}110^\circ\text{E}$, $35\text{--}45^\circ\text{N}$, left box) and eastern China ($110\text{--}122.5^\circ\text{E}$, $20\text{--}45^\circ\text{N}$, right box). The dotted areas indicate statistical significance with 95% confidence.

Supplementary Fig 3. (a) Correlation coefficients between simulated dust aerosol optical depth (AOD) at 550 nm and wind speed at 850 hPa averaged over eastern China (110–122.5°E, 20–45°N) in DJF from the 150-yr IRUN simulation. (b) Correlation coefficients between observed coarse AOD at 550 nm and wind speed at 850 hPa averaged over eastern China in DJF between 2001 and 2016. Observed AOD is from Moderate Resolution Imaging Spectroradiometer (MODIS) aerosol products of the Terra satellite. Observed wind speed is obtained from NCEP/NCAR reanalysis data. Coarse mode AOD is calculated from total AOD (which was available) minus the fine mode fraction, which is only available over the ocean.

Supplementary Fig 4. Changes in longwave aerosol direct radiative effect (unit: W m^{-2}) (e) at the top of the atmosphere and (f) the surface, respectively, between weak wind and normal conditions due to the interannual variations in dust emissions.

References

Wang, H. J., Chen, H. P. & Liu, J. P. Arctic sea ice decline intensified haze pollution in eastern China. *Atmos. Ocean. Sci. Lett.* 8(1), 1–9 (2015).

Wang, H. J. & Chen, H. P. Understanding the recent trend of haze pollution in eastern China: roles of climate change. *Atmos. Chem. Phys.*, 16, 4205-4211 (2016).

Zhang, Z. et al. Possible influence of atmospheric circulations on winter haze pollution in the Beijing–Tianjin–Hebei region, northern China. *Atmos.*

- Chem. Phys. 16, 561-571 (2016).
- Xu, X. et al. Climate modulation of the Tibetan Plateau on haze in China. Atmos. Chem. Phys. 16, 1365-1375 (2016).
- Carslaw, K. S., O. Boucher, D. V. Spracklen, G. W. Mann, J. G. L. Rae, S. Woodward, and M. Kulmala (2010), A review of natural aerosol interactions and feedbacks within the Earth system, Atmos. Chem. Phys., 10, 1701–1737.
- Takemura, T., H. Okamoto, Y. Maruyama, A. Numaguti, A. Higurashi, and T. Nakajima (2000), Global three-dimensional simulation of aerosol optical thickness distribution of various origins, J. Geophys. Res., 105, 17,853–17,873, doi:10.1029/2000JD900265.
- Ginoux, P., M. Chin, I. Tegen, J. M. Prospero, B. Holben, O. Dubovik, and S. J. Lin (2001), Sources and distributions of dust aerosols simulated with the GOCART model, J. Geophys. Res., 106, 20,255–20,273, doi:10.1029/2000JD000053.
- Liao, H., J. H. Seinfeld, P. J. Adams, and L. J. Mickley (2004), Global radiative forcing of coupled tropospheric ozone and aerosols in a unified general circulation model, J. Geophys. Res., 109, D16207, doi:10.1029/2003JD004456.
- Huneeus, N., et al. (2011), Global dust model intercomparison in AeroCom phase I, Atmos. Chem. Phys., 11, 7781–7816, doi:10.5194/acp-11-7781-2011.
- Seinfeld, J., et al. (2004), ACE-ASIA: Regional climatic and atmospheric chemical effects of Asian dust and pollution, Bull. Am. Meteorol. Soc., 85, 367–380, doi:10.1175/BAMS-85-3-367.
- Dawson, J. P., B. J. Bloomer, D. A. Winner and C. P. Weaver (2014), Understanding the meteorological drivers of U.S. particulate matter concentrations in a changing climate, Bull. Am. Meteorol. Soc., 95, 521–532, doi:10.1175/BAMS-D-12-00181.1.
- Horton, D. E., C. B. Shinner, D. Singh, and N. S. Diffenbaugh (2014), Occurrence and persistence of future atmospheric stagnation events, Nat. Clim. Change, 4(8), 698–703, doi:10.1038/nclimate2272.

Responses to Reviewer #3

Summary: The paper revealed the possible interactions between the natural dust emission and wind speed over eastern China, and discussed their roles on the increase aerosol pollution, especially winter hazy pollution in recent decades, using model simulations. I think the issue of this paper is important, and some findings in the paper favor to recognize haze pollution over eastern China thoroughly. Otherwise, I also found some issues in the paper should be revised or addressed further in order to consolidate its conclusions, so I recommend a major revision before this paper can be published.

Major comments:

1. The major flaw of this paper is that the main conclusions are obtained just basing on two sensitivity simulations, with and without inter-annual variations in dust emissions. This is not convincing to me. The relationships among the wind, aerosol pollution, and dust emission have been well revealed by many previous studies from simulations. If this study is limited at the current level, I suggest rejecting because it is just a copy study and there is no novel. Some results from the observations should be added to consolidate the conclusions. Of course, just several sites are needed if the data are limited.

Response:

It is true that impacts of wind on dust emissions (Huang et al., 2014; Guan et al., 2015; Lou et al., 2016) and other aspects of the wind-aerosol pollution relationship (Zheng et al., 2015; Li et al., 2016; Wang and Chen, 2016) have been investigated in previous studies. However, no study proposed the positive dust-wind feedback, namely that weak wind decreases dust emission and decreased dust reinforces the weakening of winds over eastern China. In addition, this is the first study that highlights the influence of natural aerosols on the wintertime anthropogenic aerosol pollution over eastern China. Therefore, we believe our study contributes a novel aspect to the existing studies of haze.

The method of using interactive and prescribed emissions has been used several times to find out the interactions between natural aerosol and climate variability (Xu et al., 2015; Lou et al., 2016; Yang et al., 2016a,b,c). Although there are uncertainties in this approach, such as in the emission parameterizations and the coarse resolution of model, they are unlikely to change our main finding that dust-wind interactions intensify aerosol pollution over eastern China.

To support our results, we included atmospheric visibility data from 346 sites over China and have now added the shortwave flux data from the Clouds and the Earth's Radiant Energy System (CERES) data set for years 2001–2016. We showed in Fig. 5a the stronger correlation between observed haze days and dust storm frequency over southern China than northern China and in Fig. 5b the strong decrease in shortwave flux over southern China during

weak wind conditions, which is in agreement with our finding that the dust-wind feedback is stronger over southern China. Please see also our response to comment 3.

2. In the experiment design (Line 228-237). The initial conditions are not clear in IRUN. This simulation was driven by the dust emission or the meteorological fields? If driven by the dust emission, is it observation or simulation? Is it open or not? If driven by the meteorological fields, which variables are needed to input? Wind or others?

Response:

The CESM model used in the IRUN simulation is a fully coupled global climate model including models of the atmosphere, land, ocean, and sea ice. The natural aerosol emission and meteorological fields are calculated online. The simulated dust emission could perturb meteorology and climate and the changes in meteorology also have feedback on dust emissions. Each variable, including dust emission and wind, has an initial value at the beginning of the simulation, then the variable changes with time to represent the internal variability of the climate system. We have added in methods that: “Dust emissions and meteorological fields are calculated online.”

The dust emission in the model is first calculated for each plant functional type (PFT); then summed up using the area-weighting to give the grid-box average, i.e.,

$$\bar{F} = \sum_j A_j F_j$$

For the j th PFT of a grid-box, the vertical flux of dust mass emission is calculated by

$$F_j = TS\alpha f_m Q_{S_j}$$

Here T is an adjustable tuning parameter, which is time and space invariant. The source erodibility factor S and the sandblasting mass efficiency α are time invariant but dependent on geographical location. f_m is the fraction of grid cell area covered by exposed bare soil suitable for dust mobilization. The horizontally saltating mass flux Q_{S_j} is calculated as

$$Q_{S_j} = \begin{cases} u_{*j} + 0.003U_{10j}^2 \left(1 - \frac{u_{*t}}{u_{*j}}\right)^2 & u_{*j} \geq u_{*t} \\ u_{*j} & u_{*j} < u_{*t} \end{cases}$$

The friction velocity u_{*j} and the 10m wind speed U_{10j} are functions of surface wind speed, boundary layer stability, and characteristics of the land surface. Therefore, the simulated dust emissions are dependence on both

location and wind speed.

3. In supplementary Figure 3, the authors showed the relationship between the MAM dust storm frequency index and DJF haze days over eastern China, and their correlation coefficient can be up to +0.64 (I think it should be -0.64???). The relationships between wind and DSFI, wind and haze days in China have been well revealed by many previous studies. The relation between the DSFI and haze days is certainly clear. The most important thing here is that the author should give out the contribution rate of the wind speed due to the natural dust emission to the increased haze days.

Response:

Yes, it should be -0.64 . But, as the reviewer mentioned, the moderate negative correlation is not sufficient to support our finding that the wind-dust feedback enhances aerosol pollution. We have added the spatial distribution of correlation coefficients between DSFI and haze days and the relative differences in surface net shortwave flux during weak wind conditions in Fig. 5, which support our findings. Through simulation, we found that the wind-dust feedback could enhance the weakening of wind speeds, especially over southern China (Fig. 3a). Therefore, we expect that this effect should have a larger impact on haze over southern China than northern China. In Fig. 5a, the haze days and DSFI show stronger negative correlation over southern China ($110\text{--}122.5^\circ\text{E}$, $30\text{--}45^\circ\text{N}$) ($R = -0.76$) than northern China ($110\text{--}122.5^\circ\text{E}$, $20\text{--}30^\circ\text{N}$) ($R = -0.47$), in agreement with this expectation. Over northern China, the surface net shortwave flux decreases during weak wind conditions (Fig. 5b), indicating aerosol increases over northern China. The decrease in shortwave flux should become weaker over southern China if the dust-wind feedback does not exist, because anthropogenic aerosols are more concentrated over northern China than southern China. However, the decrease in shortwave flux is weaker around 35°N but even stronger over southern China. These changes in surface net shortwave flux further provide observational support for the feedback effects in the simulations. While observations do not allow us to quantify the magnitude of the feedback, the different regional correlation patterns provide strong support for the existence of the wind-dust feedback at a non-negligible level. The model simulations have allowed us to quantify the contribution of these feedbacks (13% in this study).

We revised the description of DSFI and haze days as “Observed dust storm frequency index (DSFI), haze days and surface net shortwave flux are used here to support the finding that dust-wind feedback enhances aerosol pollution over eastern China. The weakening wind speed induced by the dust-wind feedback is stronger over southern China than northern China simulated in this study (Fig. 3a). The observed haze days and DSFI also show a stronger negative correlation over southern China than northern China, with correlation coefficients of -0.76 averaged over southern China and -0.47

averaged over northern China (Fig. 5a). The stronger correlation over southern China suggests that, in addition to the wind induced co-variation of dust and haze, there are other factors influence the relationship of dust and haze, especially over southern China. This correlation is consistent with our finding that dust-wind interactions intensify aerosol pollution over eastern China. In addition, over northern China, the surface net shortwave flux decreases during weak wind conditions (Fig. 5b), indicating aerosol increases over northern China. The decrease in shortwave flux should become weaker if the dust-wind feedback did not exist, because anthropogenic aerosols are more concentrated over northern China than southern China in DJF. However, the decrease in shortwave flux is weaker around 35°N but even stronger over southern China. These changes in surface net shortwave flux further provide observational support for the feedback effects in the simulations that enhance aerosol pollution over eastern China.”

4. In recent years, the impacts of climate change on aerosol pollution has been taken increasing attentions. I suggest the authors should also give a review in this aspect. I suggest some referees as follow.

Wang, H. J., H. P. Chen, and J. P. Liu, 2015: Arctic sea ice decline intensified haze pollution in eastern China. *Atmos. Oceanic Sci. Lett.*, 8(1), 1-9.

Zhang, Z., X. Zhang, D. Gong, S.-J. Kim, R. Mao, and X. Zhao, 2016: Possible influence of atmospheric circulations on winter haze pollution in the Beijing-Tian-Hebei region, northern China. *Atmos. Chem. Phys.*, 16, 561-571.

Xu, X., T. Zhao, et al., 2016: Climate modulation of the Tibetan Plateau on haze in China. *Atmos. Chem. Phys.*, 16, 1365-1375.

Wang, H. J., and H. P. Chen, 2016: Understanding the recent trend of haze pollution in eastern China: roles of climate change. *Atmos. Chem. Phys.*, 16, 4205-4211.

Response:

Thanks for the suggestion. We have added these studies in the introduction section as “In recent years, the impacts of climate change on aerosol pollution have received increasing attention (Dawson et al., 2014; Horton et al., 2014). Aerosol pollution events over eastern China have been attributed to changes in meteorological fields and climate indices associated with climate change, such as Arctic sea ice (Wang et al., 2015), local precipitation, surface wind (Wang and Chen, 2016), and atmospheric circulation (Xu et al., 2016; Zhang et al., 2016).”

Figure 5. (a) Spatial distribution of correlation coefficients between spring dust storm frequency index (DSFI) in China and DJF haze days defined as days with observed atmospheric visibility less than 10 km and relative humidity less than 90% for 1981–2007. The DSFI data are obtained from previous study defined by the leading principal components of empirical orthogonal function analysis of spring (March to May) dust storm frequency. Observed visibility data are derived from National Climatic Data Center (NCDC) Global Summary of Day (GSOD) database. (b) Spatial distribution of relative differences in surface net shortwave flux (FSNS, unit: %) between weak wind and normal conditions based on NCEP/NCAR wind speed. The shortwave flux data are derived from the Clouds and the Earth's Radiant Energy System (CERES) data set for years 2001–2016. Positive values represent net downward fluxes. The relative differences are calculated by $(V_{\text{Weak}} - V_{\text{Normal}}) / \overline{V_{\text{Normal}}}$, which remove spatial variability of the variable.

References

Zheng, G. J. et al. Exploring the severe winter haze in Beijing: the impact of

- synoptic weather, regional transport and heterogeneous reactions. *Atmos. Chem. Phys.* 15, 2969-2983 (2015).
- Li, Q., Zhang, R. & Wang, Y. Interannual variation of the wintertime fog-haze days across central and eastern China and its relation with East Asian winter monsoon. *Int. J. Climatol.* 36, 346–354 (2016).
- Wang, H. J. & Chen, H. P. Understanding the recent trend of haze pollution in eastern China: roles of climate change. *Atmos. Chem. Phys.* 16, 4205-4211 (2016).
- Huang, J. et al. Climate effects of dust aerosols over East Asian arid and semiarid regions. *J. Geophys. Res. Atmos.* 119, 11,398–11,416 (2014).
- Guan, Q. et al. Climatological analysis of dust storms in the area surrounding the Tengger Desert during 1960–2007. *Clim. Dyn.* 45, 903–913 (2015).
- Lou, S. et al. Impacts of the East Asian Monsoon on springtime dust concentrations over China. *J. Geophys. Res. Atmos.* 121, 8137–8152 (2016).
- Yang, Y., L. Russell, S. Lou, M. Lamjiri, Y. Liu, B. Singh, and S. Ghan (2016a), Changes in sea salt emissions enhance ENSO variability, *Journal of Climate*, doi:10.1175/JCLI-D-16-0237.1.
- Yang, Y., et al. (2016b), Impacts of ENSO events on cloud radiative effects in preindustrial conditions: Changes in cloud fraction and their dependence on interactive aerosol emissions and concentrations, *J. Geophys. Res. Atmos.*, 121, doi:10.1002/2015JD024503.
- Yang, Y., L. M. Russell, S. Lou, Y. Liu, B. Singh, and S. J. Ghan (2016c), Rain-aerosol relationships influenced by wind speed, *Geophys. Res. Lett.*, 43, doi:10.1002/2016GL067770.
- Lou, S., L. M. Russell, Y. Yang, L. Xu, M. A. Lamjiri, M. J. DeFlorio, A. J. Miller, S. J. Ghan, Y. Liu, and B. Singh (2016), Impacts of the East Asian Monsoon on springtime dust concentrations over China, *J. Geophys. Res. Atmos.*, 121, 8137–8152, doi:10.1002/2016JD024758.
- Xu, L., D. W. Pierce, L. M. Russell, A. J. Miller, R. C. J. Somerville, C. H. Twohy, S. J. Ghan, B. Singh, J. Yoon, and P. J. Rasch (2015), Interannual to decadal climate variability of sea salt aerosols in the coupled climate model CESM1.0, *J. Geophys. Res. Atmos.*, 120, 1502–1519, doi:10.1002/2014JD022888.

Reviewers' comments:

Reviewer #1 (Remarks to the Author):

The authors have revised the manuscript following the comments and suggestions and significantly improved the manuscript. Therefore, I recommend to accept for publication.

Reviewer #2 (Remarks to the Author):

In the second submission, the authors supplied more analysis and made a detailed response. There is no doubt that the model simulations can produce dust-wind interactions associated with air pollution in China with corresponding assumptions. However, so far, I still did not see any obvious evidence that such phenomenon or "variations" existed. If the other reviewers support publication, I suggest this "mechanism" should be taken as a scenarios simulation rather than facts existed in China. Here are some specific comments for reference:

1 The authors used correlation between MODIS coarse AOD over ocean and NCEP winds to validate their simulations which exhibited notable difference with the simulated results in supplied Figure 3. Aerosol loading over ocean can be influenced by airflows from the diverse continental emissions. Why the authors did not use MODIS Deep Blue AOD over the deserts? And what's the true trends? There are both long-term satellite and ground-based observations in northwestern China.

2 Conclusion such as "The dust-induced lower winds enhance stagnation of air and account for about 13% of increasing aerosol concentrations over eastern China" existed only in assumed scenarios. Even if dust emissions decreased by 29% as the authors assumed, the dust-induced changes can be impacted by several factors. First, different from widespread dust storms, transport of dust plumes only impacted limited areas. Also transport path and distribution varied largely in these dust events. Moreover, much larger induced effects can occur in local areas of eastern China considering the high aerosol loading. In fact, aerosol radiative effect played a relatively small role in variation of atmospheric circulation.

3 Considerable uncertainties can also exist in dust emissions, transport as well as their interactions in modeling simulation. So, the authors should be rigorous in their expression and analysis.

Reviewer #3 (Remarks to the Author):

The issues have been satisfactorily addressed and I have no further comments.

Responses to Reviewer #2

In the second submission, the authors supplied more analysis and made a detailed response. There is no doubt that the model simulations can produce dust-wind interactions associated with air pollution in China with corresponding assumptions. However, so far, I still did not see any obvious evidence that such phenomenon or “variations” existed. If the other reviewers support publication, I suggest this “mechanism” should be taken as a scenarios simulation rather than facts existed in China. Here are some specific comments for reference:

Response:

We appreciate the reviewer’s efforts to improve the quality of the manuscript. To address the reviewer’s concern about observational evidence, we have used the following ground-based observations, satellite data, and reanalysis data to support our results:

1. Daily dust storm frequency over 753 Chinese meteorological sites for 1981–2015 provided by the China Meteorological Administration to illustrate the dust variability over China and the dust–wind feedback (as described in revised manuscript and detailed responses to review). The latter more extensive data set is proprietary and so Dr. J. Guo has been added as an author. (Fig. 5a, Supplementary Fig. 3, and Supplementary Fig. 7a, added in revised manuscript)
2. Daily atmospheric visibility data collected data from 346 meteorological stations in China for 1981–2015 from National Climatic Data Center (NCDC) Global Summary of Day (GSOD) database also show evidence of the dust–wind feedback proposed in this study. (Fig. 5a and Supplementary Fig. 7a, added)
3. Surface net shortwave flux derived from the Clouds and the Earth’s Radiant Energy System (CERES) data set for years 2001–2016 also provide support for the dust–wind feedback. (Supplementary Fig. 7b)
4. Wind fields from GEOS-4 assimilated data for 1986–2006 and NCEP/NCAR reanalysis data for 1980–2016 provide an evaluation of the simulated mean and differences in wind vectors between weak wind and normal conditions, as well as the links between wind speed over the Gobi Desert and eastern China. (Supplementary Fig 1. and Supplementary Fig 2.)
5. Aerosol Index from TOMS measurements over 1979–1993 provide support for the finding that dust concentration decreases over eastern China during weak wind conditions. (Fig. 1d)

6. Aerosol optical depth (AOD) in coarse mode from MODIS/Terra over 2001–2016 supports the result that both dust emissions at the source of Gobi Desert and airborne dust over eastern China decrease during weak wind conditions. (Supplementary Fig. 4)

These various independent multi-year observational records (in particular, items 1,2,3 provide direct observational evidence of the proposed feedback), together with the simulations reported in this study, provide rather strong support for the finding that a dust-wind feedback enhances aerosol pollution over eastern China.

1 The authors used correlation between MODIS coarse AOD over ocean and NCEP winds to validate their simulations which exhibited notable difference with the simulated results in supplied Figure 3. Aerosol loading over ocean can be influenced by airflows from the diverse continental emissions. Why the authors did not use MODIS Deep Blue AOD over the deserts? And what's the true trends? There are both long-term satellite and ground-based observations in northwestern China.

Response:

Thanks for the suggestion. The dust emissions associated with strong winds are commonly produced by cyclonic activities, while increases in anthropogenic aerosol concentrations in winter are associated with unusual meteorological conditions, such as persistent anti-cyclonic circulation (Jugder and Chung, 2004). The dust variability could be masked by the strong interannual variations in anthropogenic aerosols on a monthly timescale (Jugder et al., 2011). Therefore we did not use MODIS Deep Blue AOD, but use coarse mode AOD to avoid the influence of fine particles. In addition, the deep blue algorithm measures AOD over bright surface such as deserts and arid regions with uncertainties of up to 30% (Hsu et al., 2006), which are comparable in magnitude to the 29% of changes in dust emissions over the Gobi Desert between weak wind and normal conditions in this study. In other words the uncertainties in MODIS Deep Blue AOD are too large to evaluate the dust variability in this region.

As the reviewer mentioned, aerosol loading over the ocean can be influenced by transported air masses with diverse continental emissions. However, over the downwind regions of the East China Sea, the dust from China accounts for more than 80% of the dust load in the atmosphere (Tanaka and Chiba, 2006). The differences between observed and simulated correlations probably result from the uncertainties in both model and satellite data, including emission, transport and removal parameterizations in the model, and signal uncertainty, retrieval bias, and cloud contamination in satellite data. The MODIS data used here cover only 16 years, while the model

simulates 150 years. The comparisons of the model simulations and satellite observations are therefore limited by the higher sampling error of the satellite data set.

In addition to comparing to coarse mode AOD from MODIS, we have used ground-based daily dust storm data for 1981–2015 over 753 sites in China, which are provided by the China Meteorological Administration, to verify the simulated decreased dust concentration during weak wind conditions. We have added Supplementary Fig. 3, which shows composite differences in observed dust storm frequency from 753 sites in China between weak wind and normal conditions. The surface observations show lower dust frequency over a majority of sites over both the Gobi Desert and northern China during weak wind conditions relative to normal conditions, supporting the simulated result that both dust emissions at the source of Gobi Desert and airborne dust over eastern China decrease during weak wind conditions.

We have revised the description in the manuscript as follows: “The surface dust storm observations also show less dust storm frequency over a majority of sites in both the Gobi Desert and northern China during weak wind conditions in DJF compared to normal conditions (Supplementary Fig. 3). In addition, both the simulated dust aerosol optical depth (AOD) from the 150-year IRUN simulation and the coarse AOD from Moderate Resolution Imaging Spectroradiometer (MODIS) aerosol products of the Terra satellite for 2001–2016 over the downwind regions of the East China Sea correlate positively with wind speed averaged over eastern China (Supplementary Fig. 4). However, the spatial distributions of the positive correlations show some differences between the model simulations and the satellite records, probably due to the uncertainties associated with the aerosol parameterizations in the model, cloud contamination, retrieval bias, and signal uncertainties in the satellite data. Nonetheless, these observational records support the results that both dust emissions at the source of the Gobi Desert and airborne dust over eastern China decrease during weak wind conditions.”

In this study, we showed that dust-wind interactions intensify aerosol pollution over eastern China. Given the trend toward weakening of the East Asian monsoon winds (Xu et al., 2006) and the decrease in dust days over the Gobi Desert (Hara et al., 2006) in the past few decades, our results indicate that this mechanism could be contributing to the increasing haze episodes over eastern China. We did not examine the long-term trend of dust aerosol, however, many previous studies have reported the decreasing trend of dust over the Gobi Desert (CMA, 2002; Qian et al., 2002; Wang et al., 2004; Gong et al., 2006; Hara et al., 2006).

We have also used ground-based dust and visibility observations, satellite radiative flux data, and reanalysis data to support the results in Fig. 5 and Supplementary Fig. 7. Please see also our response to comment 2.

2 Conclusion such as “The dust-induced lower winds enhance stagnation of air

and account for about 13% of increasing aerosol concentrations over eastern China” existed only in assumed scenarios. Even if dust emissions decreased by 29% as the authors assumed, the dust-induced changes can be impacted by several factors. First, different from widespread dust storms, transport of dust plumes only impacted limited areas. Also transport path and distribution varied largely in these dust events. Moreover, much larger induced effects can occur in local areas of eastern China considering the high aerosol loading. In fact, aerosol radiative effect played a relatively small role in variation of atmospheric circulation.

Response:

We agree with the reviewer that dust plumes only impact limited areas and that transport pathways and distribution vary greatly during different dust events. For this reason, we studied the dust-wind feedback on a monthly timescale. The monthly averages smooth over the event-to-event differences between individual dust storms and dust plumes, each of which has different transport pathways and distribution. The monthly averages quantify the dust aerosol influence from the perspective of climatological mean values. To make this approach more explicit, we have added a caveat in the discussion section as follows: “The results in this study use the perspective of the climatological mean by incorporating monthly-averaged model simulations. On shorter time scales, the transport pathways and spatial extent of individual dust events would be much more variable locally.”

The dust-wind feedback proposed in this study is that weak winds decrease dust emission, which reinforces the weakening of winds. For anthropogenic aerosols, the aerosol and precursor emissions are not as strongly linked to wind speed as are the emission of dust. Therefore, the large anthropogenic aerosol loading over eastern China does not have the same mechanism as the dust-wind feedback. In addition, in this study, we examine the impact of external forcing on aerosol pollution, not the impact of the anthropogenic aerosol on itself. However, anthropogenic aerosols could influence meteorological fields in other ways. For example, as we discussed in the discussion section, anthropogenic aerosols could also affect planetary boundary layer height, which also affects the concentration of aerosol pollution over China (Ding et al., 2014).

We agree that compared to the internal variability of atmospheric circulation, aerosols play a relatively small role in affecting atmospheric circulation. However, aerosol effects are still important, as they provide understanding of the mechanisms that control atmospheric circulation and its variability. Another example of this type of effect is that radiative forcing of natural aerosols was reported to account for about 70% of variations in northern tropical Atlantic Ocean temperatures (Evan et al., 2009). Based on model simulations, Evan et al. (2011a) found that the radiative effect of African dust could perturb the Atlantic Meridional Mode and influence ocean-atmosphere variability in the tropical Atlantic. Booth et al. (2012) also found

that aerosols were implicated as a prime driver of twentieth-century North Atlantic climate variability. Evan et al. (2011b) showed that anthropogenic aerosols intensify Arabian Sea tropical cyclones. Aerosols are also found to enhance storm track intensity by invigorating the intensity of tropical cyclones (Wang et al., 2014a, 2014b, 2014c). Bollasina et al. (2011) found anthropogenic aerosols play an important role in the weakening of the South Asian summer monsoon. Therefore, although aerosol effects are relatively small, they are important for understanding variability in atmospheric circulation. We have added these descriptions in the discussion section of manuscript.

To further address the reviewer's concern that our findings apply only to the specific assumptions of our simulations, we have used ground-based daily dust storm data provided by the China Meteorological Administration (CMA) together with daily atmospheric visibility data in China to compare with our results. We have added Fig. 5 showing the lagged correlation coefficients between observed daily dust storm events averaged over the Gobi Desert region and atmospheric visibility in China in DJF for years 1981–2015. The lag time for the atmospheric visibility relative to the leading dust storm events are from 0 to 5 days. Observed daily dust storm events and daily atmospheric visibility measurements in China together with the lagged correlation analysis (Fig. 5) provide support for the finding that the dust-wind feedback enhances aerosol pollution over eastern China. When lagging is not applied, dust storm occurrence over the Gobi Desert and atmospheric visibility over the dust source region and the downwind northern China are negatively correlated. This result suggests that, when dust storms occurred in association with strong winds, the visibility decreased significantly over these regions on the same day as the dust storms. One day after the dust storms the visibility increased due to the decreased dust aerosol concentration, with 61 of 154 sites over eastern China showing statistically significant positive correlations over eastern China. Lags of two and three days showed that the visibility increased substantially in eastern China, with statistically significant positive correlations at 91 and 92 sites, respectively. If the sudden increases in positive correlations at 2 to 3 days were only due to the dust-induced visibility changes, the correlations would not have large changes after three days. However, these large positive correlations decreased substantially at lags of four and five days, with only 79 sites showing statistically significant positive correlations over eastern China. This result indicates that a meteorological response to the Gobi dust variability exists with 2 to 3 day lag times, leading to decreases (increases) in visibility and increases (decreases) in aerosol concentrations over eastern China when there are lower (higher) dust emissions over the Gobi Desert. This lagged correlation analysis of daily data strongly support our finding that decreases in dust emissions reinforce weakening of wind speed and intensify aerosol pollution over eastern China.

In addition, the weaker wind speed induced by the dust-wind feedback

simulated in this study is stronger over southern China than over northern China (Fig. 3a). The observed haze days and dust storm frequency averaged over the Gobi Desert in DJF also show a stronger negative correlation over southern China than over northern China (Supplementary Fig. 7a). The decrease in surface net shortwave flux derived from the Clouds and the Earth's Radiant Energy System (CERES) is also larger over southern China during weak wind conditions than in normal wind conditions (Supplementary Fig. 7b), even though anthropogenic aerosols are more concentrated over northern China. These results also provide observational support for the contribution of the modeled feedback effects in recent observations, although the haze days and shortwave flux are also influenced by dust variability.

We have added these descriptions to the last paragraph of the results section.

3 Considerable uncertainties can also exist in dust emissions, transport as well as their interactions in modeling simulation. So, the authors should be rigorous in their expression and analysis.

Response:

We appreciate this reminder. We have carefully and rigorously revised nomenclature and wording for the entire manuscript, in order to specifically distinguish what results are “based on model simulations” or “simulated values” as opposed to observational records

We have also added a caveat in the discussion section, as “Considerable uncertainties exist in simulations of dust emissions as well as in their transport and other interactions with meteorology in modeling simulations. For this reason, different models may provide different estimates of the specific magnitude of the dust-wind feedback. However, we expect that the basic physics that drive these results will provide the same type of feedback effect in all models that include wind-dependent dust emission parameterizations and aerosol radiative forcing effects on circulation.”

Supplementary Fig 3. Composite differences in observed dust storm frequency (unit: %) over 753 sites in China between weak wind and normal conditions based on 850 hPa wind speed from NCEP/NCAR meteorological fields for 1981–2015. Dust storm frequency is defined as dust storm days per examined days. Sites without dust day in DJF are not shown.

Figure 5. Lagged correlation coefficients between observed daily dust storm

events averaged over the Gobi Desert region and atmospheric visibility over 346 meteorological stations in China in DJF for years 1981–2015. Sites with outlines indicate statistical significance with 95% confidence. The lag time for the atmospheric visibility relative to the leading dust storm events are from 0 to 5 days, as shown at the bottom right of each panel. The number of stations with statistically significant positive correlations over eastern China is shown at the top left of each panel. Records from a total of 154 stations located in eastern China were used for this analysis.

Supplementary Fig 7. (a) Correlation coefficients between dust storm frequency averaged over the Gobi Desert and haze days (defined as days with observed atmospheric visibility less than 10 km and relative humidity less than 90%) for 1981–2015. Sites with outline indicate statistical significance with 95% confidence. Observed visibility data are derived from National Climatic Data Center (NCDC) Global Summary of Day (GSOD) database. (b) Relative differences in surface net shortwave flux (FSNS, unit: %) between weak wind and normal conditions based on NCEP/NCAR wind speed. The shortwave flux data are derived from the Clouds and the Earth's Radiant Energy System

(CERES) data set for years 2001–2016. Positive values represent net downward fluxes. The relative differences are calculated by $(FSNS_{Weak} - FSNS_{Normal}) / \overline{FSNS_{Normal}}$, which remove spatial variability of the variable.

References

Jugder, D., Chung, Y.S. (2004), Anticyclones over the territory of Mongolia, *The Journal of the Korean Meteorological Society*, 40 (3), 317–329.

Jugder, D., M. Shinoda, N. Sugimoto, I. Matsui, M. Nishikawa, S.-U. Park, Y.-S. Chun, M.-S. Park (2011), Spatial and temporal variations of dust concentrations in the Gobi Desert of Mongolia, *Global Planet. Change*, 78, 14–22, doi:10.1016/j.gloplacha.2011.05.003.

Hsu, N. C., S. C. Tsay, M. D. King, and J. R. Herman (2006), Deep Blue Retrievals of Asian Aerosol Properties during ACE-Asia, *IEEE Trans. Geosci. Remote Sens.*, 44, 3180–3195.

Tanaka, T. Y., and M. Chiba (2006), A numerical study of the contributions of dust source regions to the global dust budget, *Global Planet. Change*, 52(104), 88–104, doi:10.1016/j.gloplacha.2006.02.002.

Xu, M., C.-P. Chang, C. Fu, Y. Qi, A. Robock, D. Robinson, and H. Zhang (2006), Steady decline of east Asian monsoon winds, 1969–2000: Evidence from direct ground measurements of wind speed, *J. Geophys. Res.*, 111, D24111, doi:10.1029/2006JD007337.

CMA (China Meteorological Administration), 2002. Web page in April, 2002, www.cma.gov.cn.

Qian, W., L. Quan, and S. Shi (2002), Variations of the dust storms in China and its climatic control, *Journal of Climate*, 15, 1216–1229, doi:10.1175/1520-0442(2002)015<1216:VOTDSI>2.0.CO;2

Wang, X., Z. Dong, J. Zhang, and L. Liu (2004), Modern dust storms in China: An overview, *J. Arid. Environ.*, 58, 559–574, doi:10.1016/j.jaridenv.2003.11.009.

Gong, D.-Y., R. Mao, and Y.-D. Fan (2006), East Asian dust storm and weather disturbance: Possible links to the Arctic Oscillation, *Int. J. Climatol.*, 26, 1379–1396, doi:10.1002/joc.1324.

- Hara, Y., I. Uno, and Z. Wang (2006), Long-term variation of Asian dust and related climate factors, *Atmos. Environ.*, 40, 6730–6740, doi:10.1016/j.atmosenv.2006.05.080.
- Ding, A. J., X. Huang, W. Nie, J. N. Sun, V.-M. Kerminen, T. Petäjä, H. Su, Y. F. Cheng, X.-Q. Yang, M. H. Wang, et al. (2016), Enhanced haze pollution by black carbon in megacities in China, *Geophys. Res. Lett.*, 43, 2873–2879, doi:10.1002/2016GL067745.
- Evan, A. T., G. R. Foltz, D. Zhang and D. J. Vimont (2011a) Influence of African dust on ocean-atmosphere variability in the tropical Atlantic, *Nature Geoscience*, 4, 762–765.
- Evan, A. T., J. P. Kossin, C. E. Chung and V. Ramanathan (2011b) Strengthening of Arabian Sea tropical cyclones and the South Asian atmospheric brown cloud, *Nature*, 479, 94–97.
- Evan, A. T., D. J. Vimont, R. Bennartz, J. P. Kossin and A. K. Heidinger (2009) The role of aerosols in the evolution of tropical North Atlantic Ocean temperature, *Science*, 324, 778–781.
- Wang, Y., K.-H. Lee, Y. Lin, M. Levy, and R. Zhang (2014a), Distinct effects of anthropogenic aerosols on tropical cyclones, *Nature. Clim. Change*, 4, 368–373.
- Wang, Y., R. Zhang, and R. Saravanan (2014b), Asian pollution climatically modulates mid-latitude cyclones following hierarchical modelling and observational analysis, *Nature. Comm.*, 5, 3098.
- Wang, Y., M. Wang, R. Zhang, S. J. Ghan, Y. Lin, J. Hu, B. Pan, M. Levy, J. Jiang, and M. J. Molina (2014c), Assessing the effects of anthropogenic aerosols on Pacific storm track using a multi-scale global climate model, *Proc. Natl. Acad. Sci. U.S.A.*, 111(19), 6894–6899.
- Bollasina, M. A., Y. Ming, and V. Ramaswamy (2011), Anthropogenic Aerosols and the weakening of the south Asian summer monsoon, *Science*, 334(6055), 502–505, doi:10.1126/science.1204994.
- Booth, B. B. B., N. J. Dunstone, P. R. Halloran, T. Andrews, and N. Bellouin (2012), Aerosols implicated as a prime driver of twentieth-century North Atlantic climate variability, *Nature*, 484, 228–232, doi:10.1038/nature10946.

Reviewers' comments:

Reviewer #2 (Remarks to the Author):

According to the authors' revision, there have been some improvements, especially in their expression. Despite supplement of the substantial ground observation data, there is still not adequate support for some of their conclusion. I suggest the authors change their title to 'Dust-wind interactions can intensify aerosol pollution over eastern China' or others. Some detailed comments are as follows:

1 Again, compared with the prevailing airborne dust, the few dust storms cannot represent variations of airborne dust in northwestern China.

2 I am not sure whether the authors are familiar with satellite aerosol data. There are several places need to be clarified.

First, MODIS AOD is only available for Deep Blue retrievals in desert areas. So far, the authors did not show any direct evidence that the dust loading is decreasing.

Also, accuracy of MODIS AOD is higher over ocean than over land. But uncertainties of coarse fraction of AOD are much larger, and coarse fraction may be not only dominated by dust particles.

3 More importantly, the 13% is from simulation of the ideal conditions in a yearly scale, which can overestimate or overlook many factors. Such large contribution can hardly be interpreted from current studies.

Here are typical observational and related studies for reference:

Liu, Z., et al. (2008), Airborne dust distributions over the Tibetan Plateau and surrounding areas derived from the first year of CALIPSO lidar observations, *Atmos. Chem. Phys.*, 8(16), 5045-5060.

Tao, M., L. Chen, L. Su, and J. Tao (2012), Satellite observation of regional haze pollution over the North China Plain, *J. Geophys. Res.*, 117(D12), D12203.

Yumimoto, K., and T. Takemura (2015), Long-term inverse modeling of Asian dust: Interannual variations of its emission, transport, deposition, and radiative forcing, *Journal of Geophysical Research: Atmospheres*, 120(4), 1582-1607.

Gao, Y., M. Zhang, Z. Liu, L. Wang, P. Wang, X. Xia, M. Tao, and L. Zhu (2015), Modeling the feedback between aerosol and meteorological variables in the atmospheric boundary layer during a severe fog-haze event over the North China Plain, *Atmos. Chem. Phys.*, 15(8), 4279-4295.

Responses to Reviewer #2

According to the authors' revision, there have been some improvements, especially in their expression. Despite supplement of the substantial ground observation data, there is still not adequate support for some of their conclusion. I suggest the authors change their title to 'Dust-wind interactions can intensify aerosol pollution over eastern China' or others. Some detailed comments are as follows:

Response:

We appreciate the reviewer's efforts to improve the quality of the manuscript. Following the reviewer's suggestion, we have changed the title to 'Dust-wind interactions can intensify aerosol pollution over eastern China'.

1 Again, compared with the prevailing airborne dust, the few dust storms cannot represent variations of airborne dust in northwestern China.

Response:

We agree with the reviewer that a few dust storms cannot fully represent variations of airborne dust in northwestern China. However, due to the limited availability of long-term observations in northwestern China, dust storm and dust day data have been widely used to verify spatial and temporal variations in airborne dust concentration in many previous studies (Shao et al., 2003; Zhang et al., 2004; Zhao et al., 2006, 2008; Xu et al., 2007; Zhang et al., 2009). Therefore, the decrease in observed storm frequency (Supplementary Fig. 3) could partly support the simulated decrease in dust concentrations during weak wind conditions. We have therefore revised the description, as "The surface dust storm observations also show lower dust storm frequency over a majority of sites in both the Gobi Desert and northern China during weak wind conditions in DJF compared to normal conditions (Supplementary Fig. 3), partly supporting the simulated decrease in simulated airborne dust."

2 I am not sure whether the authors are familiar with satellite aerosol data. There are several places need to be clarified.

First, MODIS AOD is only available for Deep Blue retrievals in desert areas. So far, the authors did not show any direct evidence that the dust loading is decreasing.

Also, accuracy of MODIS AOD is higher over ocean than over land. But uncertainties of coarse fraction of AOD are much larger, and coarse fraction may be not only dominated by dust particles.

Response:

Thanks for the suggestion. We show in Fig. A the correlation coefficient between DJF mean MODIS Deep Blue AOD and wind speed over the Gobi Desert over 2001–2016. It is interesting that, unlike the simulation in preindustrial condition, the observed present-day AOD does not show expected positive correlation over the Gobi Desert, but even negative over

south part of the Gobi Desert, as well as strong negative correlation over central-eastern China.

To understand the reason, we showed in Fig. B the DJF mean AOD of dust aerosol and anthropogenic aerosols (including sulfate, black carbon, primary organic matter and secondary organic aerosol), respectively, for present-day condition (2010–2014) from another simulation. In DJF, AOD over the Gobi Desert is dominated by both dust and anthropogenic aerosols, and AOD of anthropogenic aerosols is much larger than dust AOD over the southern part of the Gobi Desert. During weak wind conditions, dust emissions over the Gobi desert should decrease (Shao et al., 2003). However, the weak wind condition could further lead to less ventilation and accumulation of anthropogenic aerosols (Yang et al., 2016). The decrease in dust AOD could be masked by an increase in AOD of anthropogenic aerosols in present-day satellite data (Jugder et al., 2011). That could be why there is no significant correlation of winds and AOD over the Gobi Desert but a negative correlation over the southern part of the Gobi Desert and central-eastern China (where anthropogenic aerosol dominates) in present-day satellite data. Liu et al. (2008) also point out that MODIS blue deep AOD cannot directly distinguish dust from other aerosol species.

This explanation is supported by a simple multiple linear regression on simulated data at each grid cell in the Gobi Desert as

$$R = a \times AOD_{DUST} + b \times AOD_{ANTH} + c$$

where R is the correlation coefficient in each grid cell, $a = +1.6$, $b = -1.3$, and $c = +0.2$. The positive 'a' suggests that dust provides a positive correlation over the Gobi desert, which is consistent with our results that dust column burden changes positively with wind speed. The negative 'b' indicates less ventilation and higher concentration of anthropogenic aerosols during weak wind condition, which masks the dust variability in satellite data.

To avoid providing a possibly misleading correlation, we did not add Deep Blue AOD to this study. Also, although MODIS Deep Blue AOD has been widely used to examine spatial distribution and seasonal pattern of dust (Liu et al., 2008), no study, to our knowledge, used Deep Blue AOD to examine interannual variation of dust over the Gobi Desert. Instead, interannual variation of dust emissions from central China was constrained using MODIS coarse-mode AOD in a previous study (Yumimoto and Takemura, 2015).

We agree that uncertainties of coarse fraction of AOD are much larger, and coarse fraction may not be dominated by dust particles. We have revised the description, as "However, the spatial distributions of the positive correlations show some differences between the model simulations and satellite records, probably due to the uncertainties associated with the aerosol parameterizations in the model, cloud contamination, retrieval bias, and signal uncertainties in the satellite data. In addition, coarse AOD might not be dominated by dust particles." In addition to MODIS coarse AOD, we also used ground-based dust day observation (Supplementary Fig. 3) and TOMS

Aerosol Index (Fig. 1d) to support our result. These three observational evidences could well support the simulated dust variability in this study.

3 More importantly, the 13% is from simulation of the ideal conditions in a yearly scale, which can overestimate or overlook many factors. Such large contribution can hardly be interpreted from current studies.

Here are typical observational and related studies for reference:

Liu, Z., et al. (2008), Airborne dust distributions over the Tibetan Plateau and surrounding areas derived from the first year of CALIPSO lidar observations, *Atmos. Chem. Phys.*, 8(16), 5045-5060.

Tao, M., L. Chen, L. Su, and J. Tao (2012), Satellite observation of regional haze pollution over the North China Plain, *J. Geophys. Res.*, 117(D12), D12203.

Yumimoto, K., and T. Takemura (2015), Long-term inverse modeling of Asian dust: Interannual variations of its emission, transport, deposition, and radiative forcing, *Journal of Geophysical Research: Atmospheres*, 120(4), 1582-1607.

Gao, Y., M. Zhang, Z. Liu, L. Wang, P. Wang, X. Xia, M. Tao, and L. Zhu (2015), Modeling the feedback between aerosol and meteorological variables in the atmospheric boundary layer during a severe fog-haze event over the North China Plain, *Atmos. Chem. Phys.*, 15(8), 4279-4295.

Response:

We have to clarify that 13% calculated in this study is the 13% of the increase in $PM_{2.5}$ between weak wind and normal conditions, not the 13% of averaged $PM_{2.5}$, which may have misled the referee. Compared to other factors, such as temperature inversion, changes in atmospheric boundary layer, humidity, wind speed and direction, examined in previous studies, the contribution from dust-wind feedback is smaller but non-negligible.

We agree with the referee that the influence calculated by model simulation may be different from observation. We have added a caveat in the discussion section, as “In addition, the 13% of the total increase resulting from dust-wind interaction is from a simulation that can overestimate or overlook many factors. Although observational data in this study also imply the existence of dust-wind feedback, the magnitude of the influence may differ from that in model simulations.”

Thanks for providing additional observational references. We have carefully reviewed these studies and added them in the manuscript, as “probably due to the uncertainties associated with the aerosol parameterizations in the model, cloud contamination, retrieval bias, and signal uncertainties in the satellite data (Liu et al., 2008)”, “In addition, airborne dust was also found to directly contribute to wintertime regional haze over eastern China from satellite observations (Tao et al., 2012)”, “Interannual variations in dust emissions are large (Yumimoto and Takemura, 2015) and may influence anthropogenic aerosol pollution over eastern China through changing meteorological fields”, “Also, increasing anthropogenic aerosols can also

induce a more stable atmosphere, which leads to accumulation of air pollutants and contributes to haze formation over eastern China (Gao et al., 2015)".

Fig A. Correlation coefficients between DJF mean MODIS Deep Blue AOD and wind speed over the Gobi Desert over 2001–2016. The region boxed is used to represent the Gobi Desert (100–110°E, 35–45°N).

Fig B. Simulated DJF mean AOD of (a) dust aerosol and (b) anthropogenic aerosols (including sulfate, black carbon, primary organic matter and

secondary organic aerosol), respectively, over 2010–2014. The region boxed is used to represent the Gobi Desert (100–110°E, 35–45°N).

References

- Shao, Y., et al. (2003), Northeast Asian dust storms: Real-time numerical prediction and validation, *J. Geophys. Res.*, 108, 4691, doi:10.1029/2003JD003667.
- Zhang R., M. Wang, L. Sheng, Y. Kanai, and A. Ohta (2004), Seasonal characterization of dust days, mass concentration and dry deposition of atmospheric aerosols over Qindao, China, *China Particuology*, 2, 196–199.
- Zhao, T. L., S. L. Gong, X. Y. Zhang, J. P. Blanchet, I. G. McKendry, and Z. J. Zhou (2006), A simulated climatology of Asian dust aerosol and its trans-Pacific transport. Part I: Mean climate and validation, *J. Clim.*, 19(1), 88–103, doi:10.1175/JCLI3605.1.
- Zhao, T. L., S. L. Gong, X. Y. Zhang, and D. A. Jaffe (2008), Asian dust storm influence on North American ambient PM levels: Observational evidence and controlling factors, *Atmos. Chem. Phys.*, 8, 2717–2728, doi:10.5194/acp-8-2717-2008.
- Xu, J., S. Hou, D. Qin, S. Kang, J. Ren, and J. Ming (2007), Dust storm activity over the Tibetan Plateau recorded by a shallow ice core from the north slope of Mt. Qomolangma (Everest), Tibet-Himal region, *Geophys. Res. Lett.*, 34, L17504, doi:10.1029/2007GL030853.
- Zhang, D. F., A. S. Zakey, X. J. Gao, F. Giorgi, and F. Solmon (2009), Simulation of dust aerosol and its regional feedbacks over East Asia using a regional climate model, *Atmos. Chem. Phys.*, 9, 1095–1110, doi:10.5194/acp-9-1095-2009.
- Shao, Y., and L. Wang, A climatology of northeast Asian dust events (2003), *Meteorol. Z.*, 12, 187–196.
- Yang, Y., H. Liao, and S. Lou (2016), Increase in winter haze over eastern China in recent decades: Roles of variations in meteorological parameters and anthropogenic emissions, *J. Geophys. Res. Atmos.*, 121, doi:10.1002/2016JD025136.

- Jugder, D., M. Shinoda, N. Sugimoto, I. Matsui, M. Nishikawa, S.-U. Park, Y.-S. Chun, M.-S. Park (2011), Spatial and temporal variations of dust concentrations in the Gobi Desert of Mongolia, *Global. Planet. Change*, 78, 14–22, doi:10.1016/j.gloplacha.2011.05.003.
- Liu, Z., et al. (2008), Airborne dust distributions over the Tibetan Plateau and surrounding areas derived from the first year of CALIPSO lidar observations, *Atmos. Chem. Phys.*, 8(16), 5045-5060.
- Yumimoto, K., and T. Takemura (2015), Long-term inverse modeling of Asian dust: Interannual variations of its emission, transport, deposition, and radiative forcing, *J. Geophys. Res. Atmos.*, 120, 1582–1607, doi:10.1002/2014JD022390.
- Liu, Z., et al. (2008), Airborne dust distributions over the Tibetan Plateau and surrounding areas derived from the first year of CALIPSO lidar observations, *Atmos. Chem. Phys.*, 8(16), 5045-5060.
- Tao, M., L. Chen, L. Su, and J. Tao (2012), Satellite observation of regional haze pollution over the North China Plain, *J. Geophys. Res.*, 117(D12), D12203.
- Yumimoto, K., and T. Takemura (2015), Long-term inverse modeling of Asian dust: Interannual variations of its emission, transport, deposition, and radiative forcing, *Journal of Geophysical Research: Atmospheres*, 120(4), 1582-1607.
- Gao, Y., M. Zhang, Z. Liu, L. Wang, P. Wang, X. Xia, M. Tao, and L. Zhu (2015), Modeling the feedback between aerosol and meteorological variables in the atmospheric boundary layer during a severe fog–haze event over the North China Plain, *Atmos. Chem. Phys.*, 15(8), 4279-4295.

REVIEWERS' COMMENTS:

Reviewer #2 (Remarks to the Author):

Since the authors have supplied more analysis and made corresponding revisions, I recommend publication of this work.